# Evaluating the Biological Pump Efficiency of the Last Glacial Maximum Ocean using $\delta^{13}$C

Anne L. Morée[1], Jörg Schwinger[2], Ulysses S. Ninnemann[3], Aurich Jeltsch-Thömmes[4], Ingo Bethke[1], Christoph Heinze[1]

[1]Geophysical Institute, University of Bergen and Bjerknes Centre for Climate Research, Bergen, 5007, Norway
[2]NORCE Norwegian Research Centre and Bjerknes Centre for Climate Research, Bergen, 5838, Norway
[3]Department of Earth Science, University of Bergen and Bjerknes Centre for Climate Research, Bergen, 5007, Norway
[4]Climate and Environmental Physics, Physics Institute and Oeschger Centre for Climate Change Research, University of Bern, Bern, Switzerland

*Correspondence to*: Anne L. Morée (anne.moree@uib.no)

**Abstract.** Although both physical and biological marine changes are required to explain the 100 ppm lower atmospheric $p$CO$_2$ of the Last Glacial Maximum (LGM, ~21 ka) as compared to pre-industrial (PI) times, their exact contributions are debated. Proxies of past marine carbon cycling (such as $\delta^{13}$C) document these changes, and thus provide constraints for quantifying the drivers of long-term carbon cycle variability. This modelling study discusses the physical and biological changes in the ocean needed to simulate an LGM ocean in satisfactory agreement with proxy data, and here especially $\delta^{13}$C. We prepared a PI and LGM equilibrium simulation using the ocean model NorESM-OC with full biogeochemistry (including the carbon isotopes $\delta^{13}$C and radiocarbon) and dynamic sea ice. The modelled LGM-PI differences are evaluated against a wide range of physical and biogeochemical proxy data, and show agreement for key aspects of the physical ocean state within the data uncertainties. However, the lack of a simulated increase of regenerated nutrients for the LGM indicates that additional biogeochemical changes are required to simulate an LGM ocean in agreement with proxy data. In order to examine these changes, we explore the potential effects of different global mean biological pump efficiencies on the simulated marine biogeochemical tracer distributions. Through estimating which biological pump efficiency reduces LGM model-proxy biases the most, we estimate that the global mean biological pump efficiency increased from 38 % (PI) to up to 75 % (LGM). The drivers of such an increase in the biological pump efficiency may be both biological as well as related to circulation changes that are incompletely captured by our model - such as stronger isolation of Southern Source Waters. Last, even after considering a 75 % biological pump efficiency in the LGM ocean, a remaining model-proxy error in $\delta^{13}$C exists which is 0.07 ‰ larger than the 0.19 ‰ data uncertainty. This error indicates that additional changes in ocean dynamics are needed to simulate an LGM ocean in agreement with proxy data.

## 1 Introduction

Model and proxy reconstructions of the Last Glacial Maximum (LGM) suggest major redistributions of marine biogeochemical tracers and water masses as compared to pre-industrial (PI) times, as well as lower carbon storage in both the land biosphere and atmosphere. The culmination of these changes into a ~100 ppm lower LGM atmospheric $p$CO$_2$ concentration ($p$CO$_2^{atm}$; EPICA Project Members, 2004) has driven extensive research to identify, understand, and quantify the processes contributing to these major $p$CO$_2^{atm}$ variations (e.g., Broecker, 1982; Broecker and Peng, 1986; Heinze and Hasselmann, 1993; Heinze et al., 2016; Sigman et al., 2010; Adkins,

2013; Jeltsch-Thömmes et al., 2019; Rae et al., 2018). The oceans are of particular interest as they form the largest carbon reservoir available for atmospheric exchange on millennial timescales, and in addition need to have stored the extra carbon coming from the land biosphere and atmosphere during the LGM. Both physical (circulation, solubility) and biological processes (biological pump efficiency) likely played a role in the differences between the LGM and PI oceans, although their relative importance is under debate. Between ~25 and ~60 % is attributed to biological processes and the remainder to physical changes (Bouttes et al., 2011; Buchanan et al., 2016; Khatiwala et al., 2019; Yamamoto et al., 2019; Kobayashi et al., 2015; Stein et al., 2020; Marzocchi and Jansen, 2019).

LGM model-data comparisons provide a powerful tool to test hypotheses on glacial-interglacial changes of physical-biogeochemical ocean state, to attribute observed changes to processes, and to validate paleoclimatic model simulations. $\delta^{13}C$ of DIC (total dissolved inorganic carbon) is a particularly well-suited tracer because it reflects variations in circulation, biological production, and remineralisation in parallel and because the wealth of δ13C measurements available from the sediment core archives. This was demonstrated in previous studies on LGM model-data comparisons involving $\delta^{13}C$ (e.g., Tagliabue et al., 2009; Hesse et al., 2011; Gebbie et al., 2014; Schmittner and Somes, 2016; Menviel et al., 2017; Muglia et al., 2018; Menviel et al., 2020).

Here, we explore the marine physical and biological changes needed to simulate an LGM ocean in optimal agreement with proxy data. We use the concept of the biological pump efficiency (defined as the ability of marine organisms to consume surface ocean phosphate, or more specifically the ratio of global mean regenerated to total phosphate, Sect. 2.4) to examine its effect on LGM marine biogeochemical tracer distributions in concert with physical changes. The global mean efficiency of the biological pump is strongly and nearly linearly correlated with atmospheric $CO_2$ concentrations (Ito and Follows, 2005) and is considered a key concept to understand the atmospheric $CO_2$ drawdown potential of the ocean (Ödalen et al., 2018) through its influence on the vertical gradient of marine dissolved inorganic carbon (DIC). In our evaluation and discussion, we pay particular attention to the role of Southern Source Waters (SSW, waters originating in the Southern Ocean), which are thought to be a key component in altering ocean interior tracer distributions and glacial $pCO_2^{atm}$ drawdown (e.g., Lynch-Stieglitz et al., 2016; Schmitt et al., 2012; Moy et al., 2019; Sigman et al., 2010; Ferrari et al., 2014; Morée et al., 2018; Khatiwala et al., 2019; Rae et al., 2018; Kobayashi et al., 2015; Marzocchi and Jansen, 2019; Stein et al., 2020).

Our work represents the first LGM simulation using a forced isopycnic ocean model (NorESM-OC; Schwinger et al., 2016; Tjiputra et al., 2020), where all atmospheric forcing fields have been adjusted to represent the LGM (Sect. 2). Besides a general ocean circulation model (MICOM), NorESM-OC simulates full biogeochemistry including the $^{13}C$ and $^{14}C$ carbon isotopes (model HAMOCC), as well as dynamic sea ice (model CICE) and a prognostic box atmosphere. The simulation of the carbon isotopes is particularly useful here as they i) can be directly compared to data from sediment cores (e.g., Gebbie et al., 2015; Skinner et al., 2017), ii) are influenced by both biological and physical processes (e.g., Broecker and McGee, 2013), iii) give an indication which oceanic regions could be most relevant (Schmitt et al., 2012; Morée et al., 2018; Skinner et al., 2017) ), and, given the above, iv) are useful in model evaluation (Schmittner et al., 2013; Braconnot et al., 2012). We focus on the standardized $^{13}C/^{12}C$ carbon isotope ratio ($\delta^{13}C$; Zeebe and Wolf-Gladrow (2001)), for which relatively many LGM data are available (e.g., Peterson et al., 2014; Oliver et al., 2010). In addition, the $^{14}C/^{12}C$ carbon isotope ratio (expressed as $\Delta^{14}C$) provides the model with an age tracer (radiocarbon age), which can be used to understand water mass ventilation and circulation rates, and for comparison with reconstructed $\Delta^{14}C$ (Skinner et al., 2017;

Gebbie and Huybers, 2012). We furthermore evaluate the LGM simulation against proxy and/or model reconstructions of water mass distributions, sea surface temperature, salinity, sea ice extent, export production, vertical nutrient redistribution, $p\text{CO}_2^{\text{atm}}$, the change in marine dissolved inorganic carbon, and $\text{O}_2$ (Sect. 3.2). We apply the concept of True Oxygen Utilization (TOU; Ito et al., 2004) instead of Apparent Oxygen Utilization (AOU) and make use of the explicit simulation of preformed biogeochemical tracers in our model (Tjiputra et al., 2020). This approach makes it possible to separate physical and biogeochemical drivers of the tracer distributions more thoroughly, and accounts for the role of the air-sea carbon disequilibrium pump (Khatiwala et al., 2019). We acknowledge that without a land source of C in our simulated LGM ocean (of ~ 850 Gt C, Jeltsch-Thömmes et al., 2019), nor sediments that could alter $\text{CaCO}_3$ cycling and long-term organic matter burial (Sigman et al., 2010), we do not expect to simulate the full range of processes contributing to glacial-interglacial $p\text{CO}_2^{\text{atm}}$ changes. Rather, we include estimates of these carbon reservoir changes in our evaluation of the LGM biological pump efficiency (Sect. 3.3).

The evaluation against proxy data allows us to evaluate both the physical and biological changes needed for simulating the LGM ocean. Notably, in fully coupled paleo Earth System Modelling such as in the most recent Paleo Modelling Intercomparison Project 3 (PMIP3), only two out of nine Earth System Models included marine biogeochemistry in their LGM simulation (IPSL-CM5A-LR (Dufresne et al., 2013) and MIROC-ESM (Sueyoshi et al.,2013)). Earth System Models of intermediate complexity (and coarse resolution ocean model studies) have shown that changes in model (biogeochemical) parameterizations are needed to simulate glacial-interglacial cycles in agreement with proxy records (e.g., Jeltsch-Thömmes et al., 2019; Ganopolski and Brovkin, 2017; Buchanan et al., 2016; Heinze et al., 1991; Heinze and Hasselmann, 1993; Heinze et al., 2016). In our forced ocean model setup, we are able to reveal aspects important for modelling the LGM and relevant for improving the agreement between fully coupled paleo modelling and proxy data. Moreover, our work will help to gain insight in the changes (i.e. physical and biological) needed to simulate a different climate state (such as the LGM) - which also applies to Earth System Model-based climate projections.

## 2 Methods

### 2.1 Model description

We apply the ocean carbon-cycle stand-alone configuration of the Norwegian Earth System Model (NorESM) as described by Schwinger et al. (2016), but with several modifications for the next generation NorESM version 2 already included. The physical ocean component MICOM (Miami Isopycnic Coordinate Ocean Model; Bentsen et al., 2013) has been updated as described in Guo et al. (2019). The biogeochemistry component HAMOCC (HAMburg Ocean Carbon Cycle model; Maier-Reimer, 1993; Maier-Reimer et al., 2005) adopted for use with the isopycnic MICOM (Assmann et al., 2010; Tjiputra et al., 2013; Schwinger et al., 2016) has undergone a few minor technical improvements (e.g. updated initialisation based on latest data products, additional diagnostic tracers) and employs a new tuning of the ecosystem parameterization as described in Tjiputra et al. (2020).

In addition to these changes, the carbon isotopes ($^{13}\text{C}$ and $^{14}\text{C}$) are implemented in HAMOCC (Tjiputra et al. (2020)), a prognostic box atmosphere is made available for atmospheric $\text{CO}_2$ (including $^{13}\text{CO}_2$ and $^{14}\text{CO}_2$; Tjiputra et al., 2020), and an LGM setup is made (Sect. 2.2). This is an ocean-only modelling study, where the atmospheric forcing is prescribed from a data set (except atmospheric $\text{CO}_2$, $\delta^{13}\text{C}$ and $\Delta^{14}\text{C}$, which evolve freely; Sect. 2.2). All

simulations in this study are done without the sediment module of HAMOCC (this is done in order to avoid prohibitively long spin-up times, especially for the carbon isotopes; an acceleration method for the model spin-up including interactive water column-sediment interaction is work in progress for a separate manuscript). Applying the current set-up, detritus arriving at the sediment-water interface is evenly redistributed over the entire water

column, while opal and $CaCO_3$ are dissolved immediately in the bottom-most mass containing layer. Riverine input of carbon and nutrients is also turned off. Furthermore, nitrogen deposition, denitrification and nitrogen fixation are excluded from our simulations, as these processes cause a long-term drift in the alkalinity inventory of the ocean (and thereby the $p$CO$_2$ of the prognostic atmosphere).

The two main isotopes of carbon, $^{13}$C and $^{14}$C, are newly implemented in HAMOCC (Tjiputra et al., 2020). The

model includes fractionation during air-sea gas exchange and photosynthesis, as well as radiocarbon decay. Fractionation during $CaCO_3$ formation is small as compared to the effects of air-sea gas exchange and photosynthesis, as well as uncertain (Zeebe and Wolf-Gladrow, 2001) and is therefore omitted (e.g., Schmittner et al., 2013; Lynch-Stieglitz et al., 1995). Air-sea gas exchange fractionation (~8-11 ‰) is a function of surface ocean temperature and the $CO_3^{2-}$ fraction of total DIC such that fractionation increases with decreasing temperatures

(Zhang et al., 1995; Mook, 1986). Biological fractionation (~19 ‰) increases surface water $\delta^{13}$C of DIC while producing low-$\delta^{13}$C organic matter. In the interior ocean, this light isotope signal from organic matter is released back into the water column during remineralization and respiration, thereby creating a vertical gradient. HAMOCC applies the parameterization by Laws et al. (1997), where the biological fractionation $\varepsilon_{bio}$ depends on the ratio between phytoplankton growth rate and the aqueous $CO_2$ concentration. For $^{14}$C, each fractionation factor is set to

the quadratic of the respective $^{13}$C value (i.e., $\alpha_{14C} = \alpha^2_{13C}$). In addition, $^{14}$C is radioactive and decays with a half-life of 5730 years to $^{14}$N.

In order to evaluate the carbon isotopes against observations, we derive $\delta^{13}$C and $\Delta^{14}$C. $\delta^{13}$C is calculated using the standard equation $\delta^{13}C = \left(\frac{^{13}C/^{12}C}{(^{13}C/^{12}C)_{standard}} - 1\right) * 1000$ ‰, where $(^{13}C/^{12}C)_{standard}$ is the Pee Dee Belemnite standard ratio (0.0112372; Craig (1957)). $\Delta^{14}$C is calculated by standardizing DI$^{14}$C following $\delta^{14}C =$

$\left(\frac{^{14}C/C}{(^{14}C/C)_{standard}} - 1\right) * 1000$ ‰, where $(^{14}C/C)_{standard}$ is the NBS standard ($1.170 \cdot e^{-12}$; Orr et al., 2017). $\Delta^{14}$C is then calculated from $\delta^{14}$C, following $\Delta^{14}C = \delta^{14}C - 2 * (\delta^{13}C + 25) * (1 + \frac{\delta^{14}C}{1000})$. $\Delta^{14}$C age presented in this study is derived from $\Delta^{14}$C of DIC following $(\Delta^{14}C_{age} = -8033 * \ln(-8033 * \Delta^{14}C/1000) + 1)$ and is based on calibrated $\Delta^{14}$C of DIC using an atmospheric value of 0 ‰ for both the LGM and PI spinup (Tjiputra et al., 2020). This approach facilitates comparison with the radiocarbon disequilibrium data by Skinner et al. (2017).

**2.2 Last Glacial Maximum set-up**

Several adjustments were made to the model in order to obtain an LGM circulation field. First, the land-sea mask and ocean bathymetry were adjusted for the ~120 m lower sea level in the LGM caused by the increased land ice volume as compared to the PI. Following the PMIP4 guidelines in Kageyama et al. (2017) the Bering Strait is closed, and the Canadian Archipelago (including Borrow Strait and Nares Strait), Barents Sea, Hudson Bay, Black

Sea, Red Sea, as well as the Baltic and North Seas are defined as land in the LGM. The PI land-sea mask formed the basis for the LGM land-sea mask, through shifting the PI bathymetry 116 m upwards. If the resulting depth in a grid cell was between 0-25 meters, the depth was set to 25 m and negative depths were set to land grid points. After this, any channels with a width of only one grid cell were closed off as well, as these inhibit sea ice movement

in the sea ice model causing unrealistic sea ice build-up. LGM freshwater runoff is routed to the nearest ocean grid cell but otherwise unadjusted.

Changes in isopycnal densities and sea surface salinity restoring are applied in the LGM model setup in order to ensure an adequate vertical model resolution and ocean circulation. A net LGM increase in density due to decreased ocean temperatures and increased ocean salinity required increasing all 53 isopycnal layer densities by 1.3 kg m$^{-3}$ in the LGM setup as compared to the PI model setup. NorESM-OC uses salinity restoring to avoid long-term drift away from a predefined SSS state. Here, this predefined state is chosen, consistent with the atmospheric forcing (see below) as the mean of the LGM minus PI SSS anomaly modelled by PMIP3 models added to a PI SSS climatology. However, the unadjusted application of the PMIP3-based SSS anomaly caused an Atlantic water mass distribution and overturning strength in poor agreement with proxy reconstructions (SM 1; Fig. S2). Earlier studies have shown a high sensitivity of models to SSS restoring, especially in the North Atlantic (Rahmstorf, 1996; Spence et al., 2008; Bopp et al., 2017). Indeed, the density contrast between Northern and Southern source waters drives the simulated Atlantic Meridional Overturning Circulation (AMOC) strength in many of the PMIP2 models (Weber et al., 2007), and is therefore important for the simulation of overturning strength in agreement with proxy records. Therefore, we adjust the SSS restoring present in NorESM-OC to obtain a circulation field in better agreement with proxy reconstructions: In addition to the PMIP3-based SSS anomaly, we apply a region of -0.5 psu in the North Atlantic and +0.5 psu in the Southern Ocean (for specifics, see SM 1), as done similarly by Winguth et al. (1999) or through freshwater fluxes by Menviel et al. (2017) and Bopp et al. (2017).

An atmospheric LGM forcing for NorESM-OC was created by adding anomalies (relative to the pre-industrial state) derived from PMIP3 models (Morée and Schwinger, 2019; version 1, SM 2) to the CORE Normal Year Forcing (NYF; Large and Yeager, 2004). The use of mean PMIP/CMIP anomalies to force stand-alone models is a standard approach that has been tested before (Mitchell et al., 2017; Chowdhury and Behera, 2019; Muglia et al., 2015; Muglia et al., 2018). Through this approach, the effect of the presence of sea ice on the atmospheric state is included in the forcing, but the sea ice model handles the actual formation/melt of sea ice. Compared to the PI CORE-NYF, the LGM forcing over the ocean has a lower specific humidity (especially in the tropics), decreased downwelling longwave radiation, precipitation and air temperature, and a heterogeneous change in downwelling shortwave radiation and zonal and meridional winds. In addition to the adjustments to the NYF, the dust fluxes of Lambert et al. (2015) are used in the LGM model setup, following PMIP4 guidelines (Kageyama et al., 2017). The change to the dust forcing alters the amount of iron available for biology during photosynthesis, and is considered an important component of glacial ocean biogeochemistry and $p$CO$_2$$^{atm}$ drawdown (e.g., Yamamoto et al., 2019; Kohfeld et al., 2005; Bopp et al., 2003; Lamy et al., 2014; Ziegler et al., 2013). The PI setup uses the Mahowald et al. (2006) dust dataset. Note that atmospheric $\delta^{13}$C can freely evolve in our setup due to the inclusion of a prognostic box atmosphere.

## 2.3 Initialization and tuning

All marine biogeochemical tracers were initialized in the LGM as done for the PI spin-up using the WOA and GLODAPv2 data sets. For the LGM, consistent with the decreased ocean volume, all biogeochemical tracer concentration are increased at initialization by 3.26 %. Similarly, ocean salinity is uniformly increased by 1 psu, following PMIP recommendations. The carbon isotopes (Sect. 2.1) are only enabled after an initial spin-up of the model in order to first obtain reasonably stabilized total carbon tracer distributions. DI$^{13}$C is initialized after 1000

years using the correlation between $\delta^{13}C$ and apparent oxygen utilization (AOU) in combination with the model's DIC distribution. We applied the $\delta^{13}C$:AOU relationship of the pre-industrial Eide et al. (2017) data ($\delta^{13}C_{PI} = -0.0075 \cdot AOU + 1.72$) and converted to absolute model $^{13}C$ using model DIC and AOU. As this approach uses the model's 'native' AOU and DIC, the equilibration time of $\delta^{13}C$ was reduced as compared to initialisation with a $\delta^{13}C$ data product such as that of Eide et al. (2017). Model $DI^{14}C$ is initialized after 4000 years by first calculating $\delta^{14}C$ using a combination of pre-industrial $\delta^{13}C$ (Eide et al., 2017) (with the missing upper 200m copied from 200m depth to all empty surface layers) and the observational-based estimate of pre-industrial $\Delta^{14}C$ (Key et al., 2004). Then, model $DI^{14}C$ is derived from the $\delta^{14}C$ by rewriting and solving the standardization equation ($\delta^{14}C = \left(\frac{^{14}C/C}{(^{14}C/C)_{standard}} - 1\right) * 1000$ ‰, with model DIC as C). Subsequently, isotopic DOC, POC, phytoplankton C, and zooplankton C are initialized as done for the corresponding total carbon variable, but multiplied with 0.98 (as an estimate of the photosynthetic fractionation effect) and the respective $DI^{13}C/DI^{12}C$ or $DI^{14}C/DI^{12}C$ ratio. Isotopic $CaCO_3$ is initialized as for total carbon, multiplied with $DI^{13}C/DI^{12}C$ or $DI^{14}C/DI^{12}C$, as we do not consider fractionation during $CaCO_3$ formation.

The prognostic $pCO_2^{atm}$ is initialized at 278 ppm for both spin-ups. At initialization of the carbon isotopes, atmospheric $\delta^{13}C$ is set to -6.5 ‰ and atmospheric $\Delta^{14}C$ is set to 0 ‰, after which these are allowed to freely evolve. $pCO_2^{atm}$ at the time of initialization is then used to calculate the absolute $^{13}C$ and $^{14}C$ model concentrations ($^{13}C^{atm}$ and $^{14}C^{atm}$, in ppm).

Two main spin-ups have been made with NorESM-OC: One for the LGM and one for the PI, designed as described in Sect. 2.1-2.3. Both the PI and LGM simulations are run for a total of 5600 years.

## 2.4 Analysis of the biological pump efficiency

Here, we explore the effect of an increase in the global mean biological pump efficiency ($\overline{BP_{eff}}$, Eq. 1), which we define, following Ito and Follows (2005), as the ratio between global mean regenerated phosphate ($\overline{PO_4^{reg}}$) and global mean total phosphate ($\overline{PO_4}$).

$$\overline{BP_{eff}} = \overline{PO_4^{reg}}/\overline{PO_4} \tag{1}$$

Regenerated phosphate is calculated as the difference between total phosphate and preformed phosphate ($PO_4^{pref}$). $PO_4^{pref}$ is explicitly simulated in the model (Tjiputra et al., 2020), and represents phosphate that leaves the mixed layer in inorganic form (unutilized by biology).

We work with the global mean value of $\overline{BP_{eff}}$ as this governs $pCO_2^{atm}$ (Ito and Follows, 2005; Ödalen et al., 2018). However, we note that major local differences in the ratio of regenerated to total phosphate exist in the ocean, for example between North Atlantic Deep Water (high-ratio) and Antarctic Bottom Water (low-ratio) (Ito and Follows, 2005; DeVries et al., 2012), which thus indicate the differences in potential to sequester carbon and nutrients in the ocean interior. Here, changes in $\overline{BP_{eff}}$ are calculated to better understand the LGM redistribution of carbon between the land, atmosphere and ocean, and its effects on marine biogeochemistry (and corresponding proxy data). Our approach also allows us to give an upper estimate of the $\overline{BP_{eff}}$ of the LGM ocean.

In our approach, we explore the effects of different $\overline{BP_{eff}}$ on DIC, $O_2$ and $\delta^{13}C$ and $PO_4$ distributions in an offline framework (i.e., without running additional simulations). Simulated $\overline{BP_{eff}}$ is adjusted by adding or removing $PO_4^{reg}$ to or from the simulated $PO_4^{reg}$ ($\overline{PO_4^{reg}}_{model}$), while keeping the total $PO_4$ distribution the same. The

calculated local (i.e., grid-cell) change in $PO_4^{reg}$ ($\Delta PO_4^{reg}$) is consecutively used to estimate the effects on DIC, $O_2$ and $\delta^{13}C$ using the following relationships:

$$O_2^{new} = O_2 - \Delta PO_4^{reg} \times r_{O:P} \qquad (2)$$

$$DIC^{new} = DIC + \Delta PO_4^{reg} \times r_{C:P} \qquad (3)$$

$$\delta^{13}C^{new} = \delta^{13}C - R_{\delta^{13}C:PO_4^{reg}} \times \Delta PO_4^{reg} \qquad (4)$$

Model Redfield ratios $r_{O_2:P}$ and $r_{C:P}$ are set to 172 and 122, respectively (following Takahashi et al., 1985). $R_{\delta^{13}C:PO_4^{reg}}$ is the slope of the $\delta^{13}C:PO_4^{reg}$ relationship, which is found to be 0.67 in the model ($R^2$=0.76).

The spatial distribution of $\Delta PO_4^{reg}$ is an important consideration. We therefore explore the effect of changes in $\overline{BP_{eff}}$ on local $\Delta PO_4^{reg}$ by applying three different methods (visualized in Fig. S3): The first method (method 'add') equally distributes the mean change in $\Delta PO_4^{reg}$ over the entire ocean. The second method (method 'factor') takes into account the original distribution of $PO_4^{reg}{}_{model}$ (by first calculating the global value $\overline{\Delta PO_4^{reg}} = \overline{PO_4^{reg}}{}_{new} / \overline{PO_4^{reg}}{}_{model}$ and calculating for every grid-cell $PO_4^{reg}{}_{new} = PO_4^{reg}{}_{model} \overline{\times \Delta PO_4^{reg}}$). The third method is as the first method but only adding the extra regenerated tracers to SSW, the location of which is determined from the conservative PO tracer (method 'SSW', see Sect. 3.2 and Fig. 1b for the LGM PO tracer distribution).

It is important to note that $\overline{BP_{eff}}$ can be changed by several processes: through the soft- and hard tissue biological pumps, the solubility pump (Heinze et al., 1991; Volk and Hoffert, 1985) and by changes in the physical carbon pump (circulation/stratification of the water column).

## 2.5 The Bern3D model

In order to put our offline calculations of the biological pump efficiency as applied to NorESM-OC (i.e., Sect. 2.4) in perspective, we performed an additional analysis using results from the Bern3D v2.0s intermediate complexity model. The Bern3D model includes a 3-D geostrophic-frictional balance ocean (Edwards et al., 1998, Müller et al., 2006) coupled to a thermodynamic sea-ice component, a single-layer energy-moisture balance atmosphere (Ritz et al., 2011), and a 10-layer ocean sediment module (Tschumi et al., 2011). The Bern3D model components share the same horizontal resolution of 41×40 grid cells and the ocean has 32 logarithmically spaced depth layers (Roth et al., 2014). A four-box representation of the land biosphere (Siegenthaler and Oeschger, 1987) is used to calculate the dilution of carbon isotopic perturbations. Further information on the Bern3D model is provided in the supplementary materials (SM 3). We used the Bern3D model to estimate the LGM-PI change in the marine DIC inventory (ΔDIC) in regard to changes in four observational targets ($p$CO$_2^{atm}$, $\delta^{13}$C$^{atm}$, marine $\delta^{13}$C of DIC, and deep equatorial Pacific CO$_3^{2-}$), following the approach of Jeltsch-Thömmes et al. (2019). This is done by evaluating forcing-response relationships in seven generic carbon cycle mechanisms. These mechanisms are obtained from factorial deglacial simulations with the Bern3D model in a reduced form emulator - constrained by the observational targets. The seven generic forcings comprise changes in wind stress, air-sea gas exchange, the export rain ratio between organic and inorganic carbon, the remineralization depth of organic particles, coral growth, weathering of organic material, and carbon stocks in the land biosphere. More details on the experimental setup are given in Appendix A. ΔDIC is estimated with this approach to be ~3900 Gt C (±1σ range of [3350,4480] Gt C), based on the four given constraints. If only a single constraint of the four is applied, ΔDIC estimates are shifted by about 1000 Gt C towards lower values (see Appendix A, Fig. 7). Contributions to ΔDIC from different

carbon reservoirs that changed over the deglacial and eventually drew carbon from the ocean (atmosphere, land biosphere, coral reef regrowth, sedimentation-weathering imbalances) are discussed in Appendix A.

## 3 Results

The results presented in Sect. 3.1 and Sect. 3.2 are the annual mean climatologies over the last 30 years of the 5600-year PI and LGM spin-ups. The total integration time is long enough to eliminate most model drift at this point in time. Over the last 1000 years of the LGM spin-up simulation, ocean temperature does not show a significant trend, rather small oscillations with an amplitude of 0.015°C (Fig. S4). SSS is stable for all practical purposes with a small positive trend of about 0.004 psu per 1000 years (Fig. S4). Likewise, AMOC strength is stable, while Drake Passage transport shows a small negative trend superimposed by variations in its strength of about 2 Sv (Fig. S5). We present an evaluation of the PI (Sect. 3.1) and LGM (Sect. 3.2) spin-ups and compare the latter to proxy reconstructions, and discuss the LGM-PI changes by exploring the efficiency of the biological pump (Sect. 3.3).

### 3.1 The simulated pre-industrial ocean

The simulated pre-industrial ocean state has a maximum AMOC strength of ~18 ± 0.5 Sv north of 20° N, which compares favourably to the mean observational estimates of 17.2-18.7 Sv (Srokosz and Bryden, 2015; McCarthy et al., 2015), especially when noting the wide range of modelled AMOC strengths in similar forced ocean setups (Danabasoglu et al., 2014). The interannual variability of the simulated AMOC is small compared to observations (about ±4 Sv; Srokosz and Bryden, 2015), due to the annually repeating forcing. Drake Passage transport is simulated at ~114 Sv, lower than recent observational estimate of 173.3 ± 10.7 Sv (Donohue et al., 2016). The depth of the transition between the upper and lower Atlantic overturning cells at 30° S lies at ~2700 m, comparable to other model estimates (Weber et al., 2007; Fig. S6). Temperature biases (Figs. S7a,b and S8a,b) are generally modest (smaller than ± 1.5°C) for most of the ocean above 3000m, except for a warm bias related to a too deep tropical and subtropical thermocline and a warm bias related to a too strong Mediterranean outflow at mid-depth in the Atlantic. At depth (>3000 m) there is a widespread cold bias that originates from the Southern Ocean (too much deep mixing and associated heat loss to the atmosphere). Salinity biases (Figs. S7c,d and S8c,d) are generally small, except for a positive bias related to the too strong Mediterranean outflow at mid-depth in the Atlantic. Furthermore, the ocean is ~0.2-0.3 psu too fresh at depths over ~3 km. The mixed layer depth (MLD) is generally simulated too deep (compared to the observational estimates of De Boyer Montégut et al., 2004). In the high latitudes, winter month MLD biases in excess of 200 metres are present in our model. In low latitudes, MLD is about 20 metres too deep year-round. The simulated biogeochemistry of the PI ocean is described in more detail in Schwinger et al. (2016) although there have been some improvements due to the model updates mentioned above as described in Tjiputra et al. (2020). Some features of relevance for this study are summarized here: The spatial pattern of primary production (PP, Fig. S9) compares well with observation-based estimates with the exception of the tropical Pacific upwelling, where PP is too high, and the subtropical gyres and the coastal ocean where PP is generally too low. Because of too high PP and export in the equatorial Pacific, a far too large oxygen minimum zone (OMZ) with elevated concentrations of regenerated phosphate develops in the model (Fig. S10). Otherwise, the global nutrient concentrations are in reasonable agreement with modern observations (WOA,

Glodapv2). $\overline{BP_{eff}}$ is 38 % for the simulated PI ocean, in good agreement with observational estimates of 32-46 % (Ito and Follows, 2005; Primeau et al., 2013).

## 3.2 The simulated LGM ocean

### 3.2.1 The physical ocean state

Proxy-based reconstructions describe an LGM circulation that includes a shoaling of the upper circulation cell in the Atlantic (Glacial NADW) and expansion and slow-down of a cooler and more saline lower circulation cell (Glacial AABW; Adkins, 2013; Sigman et al., 2010; Ferrari et al., 2014). In this study, we assume these aspects of the LGM ocean to be qualitatively correct, and therefore aim for a model simulation in agreement with these features. We note that discussion continues as to the magnitude and veracity of these changes (e.g., Gebbie, 2014). Most reconstructions estimate a weakened AMOC for the LGM as compared to the PI state, although estimates vary between a 50 % weakening and an invigoration of AMOC (McManus et al., 2004; Kurahashi-Nakamura et al., 2017; Böhm et al., 2014; Lynch-Stieglitz et al., 2007; Muglia et al., 2018). The maximum overturning strength north of 20° N simulated by NorESM-OC is 15.6 Sv (~7 % weaker than simulated for our PI ocean, which we attribute to our adjustments and tuning of the salinity restoring). Higher uncertainties are involved with reconstructions of the strength of the Antarctic Circumpolar Current (ACC), with consensus leaning towards a slight invigoration (Lynch-Stieglitz et al., 2016; Lamy et al., 2015; McCave et al., 2013; Mazaud et al., 2010; Buchanan et al., 2016). We simulate a Drake Passage transport of 129 Sv (varying between ~128 and ~130 Sv, Fig. S5), which is about 13 % stronger than simulated for the PI ocean. The strengthening of the ACC in our simulation goes along with a strengthening in upwelling south of ~55 °S (not shown). The simulated transition between the Atlantic overturning cells shoals by ~350 m (Fig. S6), which falls within the uncertainty of reconstructions (Gebbie, 2014; Adkins, 2013; Oppo et al., 2018), and is not captured in most PMIP simulations (Muglia and Schmittner, 2015). The transition between the overturning cells lies well above the main bathymetric features of the Atlantic Ocean in our LGM simulation (as visible in Fig. 1 from the transition line in the conservative PO water mass tracer). This could have been an important feature of the glacial Atlantic water mass configuration due to reduced mixing along topography (Adkins, 2013; Ferrari et al., 2014) - i.e., shifting water mass boundaries away from the regions of intense internal mixing increases chemical and tracer stratification. The changes in water mass circulation cause an increased SSW volume contribution to the Atlantic and Pacific basins, as visible from the conservative PO tracer (Fig. 1a,b and Fig. S11 for the Pacific) (Broecker, 1974), and in agreement with proxy reconstructions. Furthermore, the Atlantic abyssal cell is weaker in the LGM simulation as compared to the PI, which is important for the effects of the Southern Ocean on LGM $pCO_2^{atm}$ drawdown (e.g., Kobayashi et al., 2015). Indeed, radiocarbon age increases at depth (Fig. 1c and Fig. S11 for the Pacific), with a global volume-weighted mean increase of 269 years. This is low compared to the estimate of Skinner et al. (2017) of 689±53 years, and we find the majority of our radiocarbon age bias to lie at depth in the Atlantic (not shown) indicating too strong ventilation and/or biased equilibration of these waters (which have a southern source, Fig 1a-b). Our simulation shows a radiocarbon age increase particularly at mid-depth in the South Atlantic and South Pacific (at 2-3 km; Fig. 1c and Fig. S11). Burke et al. (2015) describe this 'mid-depth radiocarbon bulge' to relate to the reorganization of the Atlantic overturning cell structure driven by Southern Ocean sea ice expansion. We simulate an increase in Southern Ocean sea ice cover for both summer and winter (Fig. S12), but less than is inferred from proxy-based reconstructions for the LGM (Gersonde et al., 2005) – similar to PMIP models (Roche

et al., 2012). If the driving mechanism of the mid-depth radiocarbon bulge is indeed expanded sea ice as argued by Burke et al. (2015), improving our simulated LGM sea ice extent (Fig. S12) may increase (mid-depth) radiocarbon ages and improve agreement with proxy records like those of Skinner et al. (2017). Besides aging of SSW (Fig. 1c), the SSW salinity increases (Fig. S13) which is in good agreement with proxy reconstructions as well (Adkins, 2013). The simulated increased SSW salinity, radiocarbon age and larger volume are important for simulation of LGM biogeochemistry and not a straightforward combination in glacial simulations (e.g., Kobayashi et al., 2015, PMIP). Nevertheless, our simulation of the LGM physical ocean state is most likely still not fully alike the LGM ocean (see also our discussion in Sect. 4). Also relevant for the water mass circulation as well as marine biogeochemistry (Marzocchi and Jansen, 2019; Jansen, 2017), are the low LGM atmospheric temperatures that cause a mean ocean temperature decrease of 1.9 °C in the model. This is less than the 2.57 ± 0.24 °C estimated from proxy reconstructions of mean ocean temperature (Bereiter et al., 2018), likely because the SSW may not carry a strong enough temperature decrease from the atmosphere into the interior ocean in our simulation (Fig. S14) – implying an underestimation of negative buoyancy fluxes. While the differences between our simulated LGM-PI changes in SST (Fig. S15) do not exhibit the same amount of heterogeneity as observed in proxy reconstructions (MARGO Project Members, 2009), the simulated mean SST change (-1.97 °C) seems reasonable when taking into account the uncertainty of SST reconstructions (MARGO Project Members (2009) estimate -1.9 ± 1.8 °C; Ho and Laepple, 2015). Further, the general pattern of stronger SST cooling outside of the polar regions is captured which is important for the air-sea disequilibrium pump (Khatiwala et al., 2019).

### 3.2.2. The biogeochemical ocean state

Proxies for the past biogeochemical state of the ocean (such as export production, oxygen concentrations, $\delta^{13}C$) allow us to make a further evaluation of our simulated LGM ocean (Fig. 2). The global mean efficiency of the biological pump $\overline{BP_{eff}}$ decreases from 38 % in the PI simulation to 33 % in the LGM simulation, as opposed to reconstructions which infer an increased regenerated signature in the interior ocean (Jaccard et al., 2009; Umling et al., 2018; Freeman et al., 2016; Yamamoto et al., 2019). The simulated increase in preformed phosphate (Fig. 2) represents an increased (but unused) potential for the ocean to draw down $p\text{CO}_2^{atm}$ (Ödalen et al., 2018). We can attribute our simulated decrease in $\overline{BP_{eff}}$ to the increase in SSW volume (Fig. 1), as SSW has a low regenerated signature (Ito and Follows, 2005). Despite the decrease in $\overline{BP_{eff}}$, simulated $p\text{CO}_2^{atm}$ is 20.3 ppm lower in our LGM setup as compared to the PI setup. We attribute this change to the net effect of the i) smaller ocean volume, causing the concentration of alkalinity, DIC and salinity and a reportedly ~16 ppm $p\text{CO}_2^{atm}$ increase (Sarmiento and Gruber, 2006), and ii) the decrease in water temperature, which drives a $p\text{CO}_2^{atm}$ drawdown of ~30 ppm (Sigman and Boyle, 2000). As we made no additional changes to the marine biogeochemical model (except for the LGM dust input field), and have no sediment or land model included in our simulation, the ~20 ppm $p\text{CO}_2^{atm}$ drawdown as well as limited changes in regenerated nutrient inventories is expected and found in earlier studies (e.g., Buchanan et al., 2016). Physical processes thus mostly drive the simulated changes in $p\text{CO}_2^{atm}$ and marine $\delta^{13}C$ between our LGM-PI simulations, also evidenced by the LGM increase in preformed DIC (Fig. S16). The atmospheric and marine mean $\delta^{13}C$ change due to glacial land-vegetation loss is not simulated because we only simulate the ocean. Simulated atmospheric $\delta^{13}C$ remains approximately constant with a small increase of 0.07 ‰ from the PI (-7.46 ‰) to the LGM (-7.39 ‰). Mean marine $\delta^{13}C$ increases 0.21 ‰ from 0.54 ‰ (PI) to 0.76 ‰

(LGM) at odds with the reconstructed LGM-PI $0.34 \pm 0.19$ ‰ decrease reconstructed by for example Peterson et al. (2014) - but partly expected in absence of a sediment or land model (see also the discussion in Sect 4.4).

Simulated changes in Atlantic total phosphate (Fig. 2) agree well qualitatively with reconstructed nutrient redistributions, which describe a deep ocean nutrient increase and mid-to surface decrease (Buchanan et al., 2016; Gebbie, 2014; Marchitto and Broecker, 2006; Oppo et al., 2018). North Pacific waters >2.5 km depth exhibit a lower LGM phosphate (and DIC) as compared to the PI, due to the lack of accumulated regenerated phosphate (Fig. S17). In agreement with the expectation of increased interior carbon storage, simulated interior DIC increases - especially in SSW (Fig. 2). As for phosphate, this increase is driven by the physical carbon pump only, through higher saturation of surface DIC in the Southern Ocean driven by lower T and increased alkalinity (see also Sect. 4 for a discussion on the contribution from the different DIC components).

However, the biases in simulated $O_2$ and $\delta^{13}C$ LGM-PI changes and their respective proxy reconstructions are large and in disagreement with proxy data (Fig. 2). Any mismatch in the absolute values of $\delta^{13}C$ is not shown here because we compare LGM-PI differences in both the sediment cores and model. In line with decreased remineralisation and increased $O_2$ solubility due to lower temperatures in the model, $O_2$ concentrations increase throughout the ocean (Fig. 2). There is a notable difference between Northern and Southern end-members in the Atlantic: Northern-source deep waters have increased $O_2$ concentrations due to physical $O_2$ pumping (colder waters have higher $O_2$ solubility) as visible in preformed $O_2$, while SSW $O_2$ increases due to a lack of remineralisation at depth due to low oxygen utilization (Fig. 3). As a last comparison, we evaluate our modelled changes in export production against proxy data, even though such data have poor global coverage and large spatial heterogeneity, and are largely qualitative (Kohfeld et al., 2005). LGM export production generally decreases outside of upwelling zones in our model (Fig. 4) and increases in upwelling zones (model and proxy data, Fig. 4). We conclude that the simulated export production increase may be too weak in the sub-Antarctic and is lacking in the tropical Atlantic (Fig. 4).

**3.3 The potential of the biological pump**

A decrease in $p$CO$_2$<sup>atm</sup> and increase in regenerated nutrients, despite an increase in low regenerated nutrient SSW volume, is likely to occur through the increase of $BP_{eff}$ (and thus regenerated nutrients) of SSW (Jaccard et al., 2009). In addition, an increase in the (Southern Ocean) air-sea disequilibrium of DIC may have kept more carbon sequestered in the deep ocean, through increased stratification and inhibition of air-sea gas exchange (Khatiwala et al., 2019) by for example sea ice (Marzocchi and Jansen, 2019). An increase in the regenerated signature of northern source water would have further contributed to a global increase in regenerated carbon and nutrients in the interior ocean (Yu et al., 2019), although occupying a smaller volume. As natural preformed concentrations in SSW are high (Ito and Follows, 2005), there is a high potential for these waters to obtain a stronger regenerated signature, and thereby facilitate $p$CO$_2$<sup>atm</sup> drawdown (Ödalen et al., 2018). Our model results for the LGM-PI change in $O_2$ and $\delta^{13}C$ show a strong mismatch with proxy records (Sect. 3.2.2 and Fig. 2). Here, we explore the effect of a potential increase of regenerated nutrients in the ocean, by increasing regenerated phosphate and updating $O_2$, DIC and $\delta^{13}C$ accordingly (Sect. 2.4). The increase is projected on the same simulated physical ocean state presumed to represent LGM conditions (i.e. Sect 3.2.1). To the extent that this physical state represents true glacial conditions (see also the discussion in Sect. 4.1), it allows for an assessment of the magnitude and nature of marine biogeochemical changes needed for lowering LGM $p$CO$_2$<sup>atm</sup>. In our approach, the additional regenerated phosphate

is taken from preformed phosphate, thus leaving the total phosphate inventory unchanged. Proxy reconstructions of global LGM phosphate, however, show that LGM phosphate was redistributed as well as elevated (Tamburini and Föllmi, 2009; Filippelli et al., 2007). As we consider a closed system (no sediments or land input of phosphate or other elements), only redistributions of phosphate can be captured in our model setup.

We compare the mean error between the model and the $\delta^{13}$C proxy data across a wide range of $\overline{BP_{eff}}$ (20-100 %, Fig. 5), and for the different methods of adding regenerated $\delta^{13}$C (Sect. 2.4). The best fit between the modelled and sediment core LGM-PI changes in $\delta^{13}$C is found for a $\overline{BP_{eff}}$ of ~ 75 % (Fig. 5). A $\overline{BP_{eff}}$ of ~ 75 % would lead to the adjusted tracer distributions shown in Fig. 6 (applying Eq. 2-4). This is true for the approach 'factor' (described in Sect. 2.4), indicating that taking the distribution of the original simulated LGM $PO_4^{reg}$ and

strengthening that regenerated signal gives the best agreement with sediment core data. The $\overline{BP_{eff}}$ of 75 % corresponds to an LGM-PI $\Delta$DIC of ~1850 Gt C. This $\Delta$DIC estimate falls below the overall range (2400 to 5500 Gt C) given by the Bern3D model and its constraints (Sect. 2.5). If using only one of the Bern3D model constraints ($p$CO$_2$^atm, $\delta^{13}$C^atm, marine $\delta^{13}$C of DIC, and deep equatorial Pacific $CO_3^{2-}$), however, the $\Delta$DIC estimate of ~1850 Gt C lies in the range of Bern3D results (see Appendix A for details). The estimated $\Delta$DIC of ~1850 Gt C allows

for full (i.e. ~80 ppm more than simulated, which is ~170 Gt C) LGM $p$CO$_2$^atm drawdown, a profound decrease in land carbon (~850 Gt C as estimated in Jeltsch-Thömmes et al., 2019) as well as a source of DIC from the deep ocean sediments/CaCO$_3$. We can however not determine these sources and sinks in our model setup.

In addition to evaluating the model-data fit of $\delta^{13}$C, we evaluate the effect of the tracer adjustments on O$_2$. The decrease in O$_2$ for an adjusted $\overline{BP_{eff}}$ (Fig. 6) shows better agreement with qualitative proxy data in the Atlantic

(Compare Figs. 2 and 6; Jaccard and Galbraith, 2011) as well as the estimated 175±20 µmol kg$^{-1}$ LGM-PI decrease in Sub-Antarctic Atlantic bottom-water (Gottschalk et al., 2016a). Absolute LGM O$_2$ (Fig. S18) would decrease to sub-zero O$_2$ concentrations in the North Pacific (ca. -100 µmol kg$^{-1}$). This is of a similar magnitude as the size of the PI model-observation bias in this area (Tjiputra et al., 2020), but may be too extreme as indicated by qualitative proxy data that describe an LGM-PI O$_2$ increase for the North Pacific above 3 km depth (Fig. S17; Jaccard and Galbraith, 2011). An increase in denitrification could have played a role here, but cannot be evaluated

in our model setup. Additional (quantitative) estimates of LGM-PI O$_2$ for major water masses would thus help to provide further constraints on LGM $\overline{BP_{eff}}$.

Our mean absolute error in $\delta^{13}$C decreases from 0.67 ‰ for the original LGM state estimate (Fig. 2) to 0.26 ‰ for the 75 % $\overline{BP_{eff}}$ ocean (Fig. 5 and 6). The remaining absolute $\delta^{13}$C error is therefore 0.07 ‰ larger than the proxy

data uncertainty. After adjustment of $\overline{BP_{eff}}$ to 75 %, the remaining mean model-data $\delta^{13}$C mismatch of 0.07 ‰ (Fig. 5) as well as the possibly too low Pacific O$_2$ (Fig. S18) indicate that projecting changes in $\overline{BP_{eff}}$ onto the simulated glacial circulation field still does not fully align with the actual LGM state - despite exploring different approaches for the redistribution of the regenerated nutrients (Sect. 2.4). In Sect. 4.4 we discuss possible mechanisms that could contribute to understand this remaining mismatch.

**4 LGM model-data biases**

Earth System Models are generally found to incompletely capture the biogeochemistry and strengthening of the biological pump for the LGM ocean, and identification of the exact processes that are missing in these models is a major challenge in modelling the LGM ocean (e.g., Galbraith and Skinner, 2020). Previous studies show that

both physical and biological marine changes must have contributed to the LGM $p$CO$_2^{atm}$ drawdown (Bouttes et al., 2011; Buchanan et al., 2016; Khatiwala et al., 2019; Yamamoto et al., 2019; Kobayashi et al., 2015; Stein et al., 2020; Marzocchi and Jansen, 2019). The physical ocean state of our LGM simulation compares quite well to proxy data (Sect. 3.2.1), while the simulated biogeochemical state (Sect. 3.2.2) reveals major disagreements with proxy data and a too weak LGM $\overline{BP_{eff}}$. Nevertheless, our modelled $\overline{BP_{eff}}$ can be increased through both physical and biological mechanisms.

Previous studies on LGM model-data comparisons employing $\delta^{13}$C of DIC reveal that a change in ocean circulation is mandatory in explaining the change in global ocean $\delta^{13}$C pattern with respect to the late Holocene and pre-industrial (Heinze et al., 1991; Winguth et al., 1999; Crucifix, 2005; Tagliabue et al., 2009; Hesse et al., 2011; Gebbie et al., 2014; Schmittner and Soames, 2016; Menviel et al., 2017; Muglia et al., 2018; Menviel et al., 2020). Some of the model simulations can achieve a fairly good fit to $\delta^{13}$C from the LGM sedimentary $\delta^{13}$C archive by varying the oceanic forcing and the ocean circulation alone (e.g., Winguth et al., 1999; Hesse et al., 2011, Gebbie, 2014; Menviel et al., 2017). However, there is no convincing case, where, alone based on circulation changes, a quantitatively correct drawdown of atmospheric $p$CO$_2$ could be explained. This dilemma had been addressed in Tagliabue et al. (2009) through stating that ocean circulation changes can explain a large part of the $\delta^{13}$C signal in the ocean, but not the atmospheric $p$CO$_2$ drawdown. The latter may require further processes - in addition to ocean circulation changes - such as an ecologically or biogeochemically induced vertical redistribution of nutrients and carbon as addressed in this study here. This viewpoint is corroborated by multi-tracer-multi-parameter studies (e.g., Heinze and Hasselmann, 1993; Brovkin et al., 2007; Heinze et al., 2016).

**4.1 Physical model-data biases of the LGM simulation**

To start, our results may indicate that the water mass production processes in the model are not (yet) fully adequate and contribute to the too low simulated LGM $\overline{BP_{eff}}$ and too weak chemocline. Interior $\delta^{13}$C is influenced by the source of deep waters as well as intermediate waters, the extent of deep convection as well as mixing processes between interior water masses (e.g., Duplessy et al., 1988), partly on spatial and temporal scales possibly not resolved by our model. An example is the role of shelf processes and the related localized sea ice formation and brine excretion in the Southern Ocean, which are highly relevant for SSW characteristics as well as the transition between the overturning cells (Klockmann et al., 2016; Mariotti et al., 2016; Bouttes et al., 2016; Jansen, 2017; Marzocchi and Jansen, 2017; Stein et al., 2020). The PI simulation of a generally too deep Southern Ocean mixed layer depth as well as Southern Ocean-attributed model biases in the PI biogeochemical tracers (Sect. 3.1; Tjiputra et al., 2020) suggest that deep water formation processes indeed are not simulated in full agreement with observational data – which may lead to biases in the LGM simulation as well. Furthermore, the lack of a reliable glacial freshwater forcing is likely to be partly responsible for biases in the LGM simulation. The weak LGM-PI reduction of SSW temperature (Fig. S14) is in disagreement with proxy records and implies an underestimation of negative buoyancy fluxes. Improving this bias may however lead to more vigorous sinking of SSW in the Southern Ocean and possible speed-up of the abyssal overturning cell, equally in disagreement with proxy data. Additionally, too strong exchange (mixing) in the LGM ocean between the deep and intermediate waters could contribute to maintain a too weak glacial chemocline. Our simulated SSW radiocarbon ages are too young compared to reconstructions (Fig. 1), consistent with inadequate isolation of these waters in the simulation. Further aging of these SSW could increase their regenerated signature (consuming O$_2$, and decreasing $\delta^{13}$C of DIC) and

steepen the chemocline while additionally improving agreement with the Bern3D model ΔDIC estimate and radiocarbon age. Indeed, both increased stratification and enhanced biological pumping in the glacial Southern Ocean likely played a role in deoxygenating SSW (Yamamoto et al., 2019).

Several key physical processes may be central to improving our model biases: i) polar wind stress forcing, related to the ice sheets (e.g., Muglia and Schmittner, 2015; Galbraith and Lavergne, 2019), ii) polar sea surface salinity forcing (this study, Rahmstorf, 1996; Spence et al., 2008; Bopp et al., 2017; Weber et al., 2007) and iii) Southern Ocean sea ice cover. The simulation of a larger increase in LGM Southern Ocean sea ice extent (Fig. S12) and/or formation rate may lead to a more stratified and isolated SSW (Ferrari et al., 2014; Nadeau et al., 2019; Jansen, 2017; Marzocchi and Jansen, 2017 and 2019; Stephens and Keeling, 2000; Stein et al., 2020). This could create older waters with higher regenerated signatures and increase the air-sea disequilibrium of biogeochemical tracers such as DIC and $O_2$ (Gottschalk et al., 2016a; Khatiwala et al., 2019; Yamamoto et al., 2019), thereby further lowering atmospheric $CO_2$ (Khatiwala et al., 2019) as well as increasing SSW radiocarbon aging.

**4.2 Biogeochemical model-data biases of the LGM simulation**

Regarding biological mechanisms that could underlie our model bias, it is instructive to first look at our simulation of a simultaneous decrease in $\overline{BP_{eff}}$ and $p\mathrm{CO_2}^{\mathrm{atm}}$ (Sect. 3.2 and 4.1). This apparent contradictive result can be attributed to the fact that our model mostly captures the physical carbon pump, as driven by temperature and ocean volume decrease. Indeed, decomposition of DIC into its components ($\mathrm{DIC^{soft}}$, $\mathrm{DIC^{pref}}$, $\mathrm{DIC^{sat}}$, $\mathrm{DIC^{bio}}$, $\mathrm{DIC^{carb}}$, and $\mathrm{DIC^{diss}}$: Fig. S16; Bernardello et al., 2014) shows that the increased ocean inventory of DIC (which is driving down $p\mathrm{CO_2}^{\mathrm{atm}}$) can be attributed to the increase in the physical $\mathrm{DIC^{pref}}$ component. Specifically, $\mathrm{DIC^{pref}}$ increases due to increased DIC disequilibrium ($\mathrm{DIC^{diss}}$) of SSW with the atmosphere as well as a strengthened solubility pump as evidenced by increased $\mathrm{DIC^{sat}}$. The more isolated abyssal cell (and accompanying increase in $\mathrm{DIC^{diss}}$) is central for the lower glacial $p\mathrm{CO_2}^{\mathrm{atm}}$ as described throughout this manuscript. Nevertheless, our study shows that the increased $\mathrm{DIC^{diss}}$ is not sufficient for the simulation of a full glacial $p\mathrm{CO_2}^{\mathrm{atm}}$ drawdown, and additional changes in the biological carbon pump must have played a role. In our simulation, $\mathrm{DIC^{soft}}$ lowers the ocean inventory of DIC in line with the simulated decrease in $\overline{BP_{eff}}$ but in disagreement with proxy records.

The lack of an increased marine C inventory through the increase in $\mathrm{DIC^{soft}}$ is also evident from simulated $O_2$ and $\delta^{13}C$. In our simulation, $O_2$ concentrations increase due to the lack of biological $O_2$ consumption at depth in SSW and increased solubility in Northern Source Waters (Fig. 3). Such an $O_2$ increase is in disagreement with proxy reconstructions (Jaccard and Galbraith, 2011) and argues for biological processes to explain the missing regenerated nutrients (i.e., remineralization) and low $\mathrm{DIC^{soft}}$ in our simulation. As for $O_2$, the widspread increase in simulated $\delta^{13}C$ of DIC contradicts $\delta^{13}C$ records from sediment cores, in which the strengthening of the vertical gradient is a main feature (Fig. 2). Deep $\delta^{13}C$ and its vertical gradient is for an important part governed by biological processes (Morée et al., 2018), further arguing for biological processes to contribute to our simulated biases. Nevertheless, $\delta^{13}C$ of DIC is governed by both ocean circulation (ventilation rate) and the efficiency of the biological pump (respiration rate), and their relative importance depends on location in complex and highly spatially heterogeneous ways (Gruber et al., 1999; Schmittner et al., 2013; Eide et al., 2017; Morée et al., 2018). A possible candidate to increase deep $\mathrm{DIC^{soft}}$ is found in Southern Ocean export production, which has a large potential to affect interior and lower latitude nutrient and DIC concentrations (Sarmiento et al., 2004; Primeau et al., 2013). Iron fertilization likely played a central role for the increase in production and export fluxes as well

(e.g., Winckler et al., 2016; Costa et al., 2016; Oka et al., 2011; Lambert et al., 2015; Yamamoto et al., 2019), and may be insufficiently captured by our model. Comparison between our simulation and proxy data point to an underestimation of LGM export production in the ice-free Southern Ocean (Fig. 4), which we conclude may have contributed to the simulated low $\overline{BP_{eff}}$ and low-regenerated signature in the biogeochemical tracer distributions.

The biological pump efficiency and hence the $\Delta\delta^{13}C$ further depend on the details of the particle flux mode through the water column. This flux depends on the size of the sinking particles, their composition, aggregation, and weight. The plankton community structure plays an important role here (Bach et al., 2019). The vertical particle flux can be accelerated, e.g., through enhanced dust deposition (clay), biogenic opal, and calcium carbonate (Haake and Ittekkot, 1990; Armstrong et al., 2002; Klaas and Archer, 2002; Honda and Watanabe, 2010; Mendes and Thomsen, 2012). The production of transparent exopolymer particles (TEP) also modifies the vertical flux mode of organic matter substantially (De la Rocha and Passow, 2007; Mari et al., 2017). These details on marine particle fluxes still need to be accounted for in marine biogeochemical models and can have differential effects on the regional biological pump efficiency. Last, we note that many of the biogeochemical processes mentioned here are closely related to ocean circulation. Therefore, changes in LGM-PI water mass configuration and overturning strength beyond those captured by our model are strong candidates for reducing model-proxy data biases.

**4.3 Other sources of model-data bias of the LGM simulation**

Besides alleviating model biases in physical or biological processes, our simulation of the LGM ocean may be improved through model development: For example, the results of our simulations on $p\mathrm{CO_2}^{\mathrm{atm}}$ may be amplified through carbonate compensation in interplay with Southern Ocean stratification (Kobayashi and Oka, 2018), if we would include the ocean sediments in our simulations. The latter would also facilitate direct comparison with sediment core records, especially if the carbon isotopes are included in the sediment model. Also, changes in the $\mathrm{CaCO_3}$ counter pump provide a "wild card" for adding to the atmospheric $p\mathrm{CO_2}$ reduction with only negligible influence on the oceanic $\delta^{13}C$ distribution. In order to reduce the uncertainties in ocean forcing during glacial times, it is vital to narrow down the freshwater flux uncertainties, e.g. through data assimilation methods. Last, a decreased $\delta^{13}C$ of DIC is possible without drastic increases in regenerated nutrients when considering changes in C:P ratios (Ödalen et al., 2020). Such C:P changes could contribute to resolve the model-proxy error and provide an interesting direction for future work.

Despite the possible shortcomings of our simulation discussed here, our simulation of a salinity increase (Fig. S13), a slowdown (Fig. S6), cooling (Fig. S14), increased volume (Fig. 1a,b) and aging (Fig. S11 and Fig. 1c) of the abyssal overturning cell combined with reduced $p\mathrm{CO_2}^{\mathrm{atm}}$ are rarely obtained (compare to e.g., Kobayashi et al., 2015; PMIP - Weber et al., 2007; Muglia and Schmittner, 2015; Sigman et al., 2010; Klockmann et al., 2016).

**4.4 The remaining model-data bias after $\overline{BP_{eff}}$ adjustment**

Changes between preindustrial and LGM ocean circulation fields as simulated by ocean models generally fail to account for the full 100-120 ppm drawdown in atmospheric $\mathrm{CO_2}$ (taken the outgassing by the land biosphere into account) when used in global ocean carbon cycle models (Heinze et al., 1991; Brovkin et al., 2007). Indeed, our study gives an estimate of ~20 ppm drawdown with an unadjusted biological pump efficiency.

The adjustment of the $\overline{BP_{eff}}$ to 75 % in addition to the reorganised circulation captures most of the magnitude of the LGM-PI $\delta^{13}C$ change (Fig. 5, 6 and S17). However, the strength of the glacial chemocline (the vertical $\delta^{13}C$

gradient) remains too weak (Fig. 6). Other modelling studies of the glacial ocean show similar biases (e.g., Heinze et al., 2016; Schmittner and Somes, 2016). This suggests that additional processes, which would allow stronger chemical stratification between intermediate and deep waters, are missing in the model(s) and are not explained in a simple way by intensification of the (soft-tissue) biological pump or our simulated changes in circulation. We recognize that optimizing the model $\overline{BP_{eff}}$ to additional (quantitative) proxies such as the nitrogen isotopes could provide more constraints (Schmittner and Somes, 2016). Besides that, our offline calculations to increase the biological pump efficiency may be too simplified to capture the vertical redistribution of regenerated nutrients – also suggested by the too weak chemocline.

Using their idealized model setup, Schmittner and Somes (2016) estimated an LGM $\overline{BP_{eff}}$ of 54-59 % (41 % in PI) and a ΔDIC of 590-790 Gt C by exploring the effects of a uniform change in the maximum growth rate of phytoplankton. A direct comparison between these studies is complicated by the large differences between the models, but the differences indicate remaining uncertainties in the magnitude of both ΔDIC and LGM $\overline{BP_{eff}}$. Nevertheless, LGM $\overline{BP_{eff}}$ seems to have been about 18 (i.e. 59-41) % and up to 37 (i.e. 75-38) % higher than in the PI. Last, we note that the total marine ΔDIC (~1850 Gt C) estimate points towards a central role for the Pacific basin to store glacial carbon, if we consider Atlantic LGM storage >3 km depth to be in the order of 50 Gt C (Yu et al., 2016).

Last, we wish to make an estimate of the effect of land biosphere and sedimentation-weathering imbalances on our analysis. The net excess transfer of light $^{12}$C to the ocean during the LGM is estimated in the Bern3D model to cause a relatively uniform ~0.4 ‰ decrease in LGM mean marine $\delta^{13}$C as compared to the PI ocean for a 890 Gt C change in land biosphere carbon, along with a similar -0.4 ‰ shift in $\delta^{13}$C$^{atm}$ (Jeltsch-Thömmes et al., 2019). The proxy-based mean marine LGM-PI $\delta^{13}$C shift of 0.34 ± 0.19 ‰ (Peterson et al., 2014) constitutes the combined effect of land biosphere, atmospheric and sedimentation-weathering imbalances in addition to a strengthening of the $\delta^{13}$C gradient (the latter of which namely represents a low deep ocean $\delta^{13}$C, thereby lowering mean marine LGM $\delta^{13}$C through its large volume). We provide an estimate of the potential effect of a shift in mean marine $\delta^{13}$C on our analysis: If we choose to make a uniform correction to the simulated LGM $\delta^{13}$C of -0.4 ‰ (such that mean marine $\delta^{13}$C LGM-PI = ~ -0.2 ‰ in our model) before comparing to proxy data, Fig. 5 looks like shown in Fig. S19a. In this case, adjusted $\overline{BP_{eff}}$ with the lowest mean absolute error is ~55 %, and corresponds to a ΔDIC of ~1000 Gt C. If we make a smaller mean shift of -0.2 ‰, we obtain a best-fit for $\overline{BP_{eff}}$ of ~65 % and a ΔDIC of ~1425 Gt C (Fig. S19b). We note, however, that suchs mean shifts in marine $\delta^{13}$C would also affect $\delta^{13}$C$^{atm}$, thus calling for the consideration of additional processes in order to fulfill the observational constraints in $\delta^{13}$C in the ocean and atmosphere simultaneously (see also discussion in Jeltsch-Thömmes et al., 2019). To our knowledge, no consensus exists on the relative importance of the different drivers on LGM-PI shifts in mean marine $\delta^{13}$C due to the large uncertainties of the LGM-PI changes in the relevant carbon reservoirs (as illustrated for the Bern3D model in Appendix Fig. 7b). We therefore limit ourselves to quantifying the potential effects of such a shift on our analysis.

## 5 Summary and conclusions

We present a model simulation of the pre-industrial and Last Glacial Maximum (LGM) oceans. We use the simulations to explore a realization of physical and biological marine changes needed to simulate an LGM ocean

in satisfactory agreement with proxy data. Despite the good agreement between (qualitative and quantitative) proxy reconstructions and our simulation of different LGM and pre-industrial (PI) ocean circulation, our model is unable to simulate the complete set of biogeochemical changes implied by proxy data. Therefore, our results (mainly the lack of increased regenerated nutrients) confirm the idea that both biogeochemical (beyond those represented by the model) and physical changes must have been at play in the ocean to create the LGM $pCO_2^{atm}$ drawdown (Heinze et al., 2016; Bouttes et al., 2011; Buchanan et al., 2016). Comparison between a range of qualitative and quantitative proxy data and simulated biogeochemical tracers (specifically total dissolved inorganic carbon, regenerated phosphate, True Oxygen Utilization, $O_2$ and $\delta^{13}C$) reveals that there is a too small signature of regenerated nutrients in our simulated LGM ocean. We conduct offline calculations of changes in the global mean efficiency of the biological pump and quantify its effect on the global mean absolute error between simulated $\delta^{13}C$ and proxy $\delta^{13}C$ data. The smallest error is found for an approximate doubling in the global mean efficiency of the biological pump $\overline{BP_{eff}}$ (from 38 % in the PI ocean to ~75 % in the LGM), when omitting the influence from land biosphere and sedimentation-weathering imbalances on mean marine $\delta^{13}C$ from our analysis. This approximate doubling is likely driven by a combination of additional biological and physical changes, such as stronger isolation of SSW (as discussed in detail in Sect 4.1 and 4.2). A $\overline{BP_{eff}}$ of ~75 % reduces the simulated global mean absolute error for $\delta^{13}C$ from 0.67 ‰ (for the originally simulated 33 % $\overline{BP_{eff}}$ of the LGM) to 0.26 ‰ - only distinguishable by 0.07 ‰ from the $\delta^{13}C$ data uncertainty. It additionally improves the agreement with both qualitative and quantitative $O_2$ reconstructions.

A mean shift of marine LGM $\delta^{13}C$ due to sedimentation-weathering imbalances and land biosphere release of low $\delta^{13}C$ can affect our analysis (Fig. S19). Our ocean model does not capture such a shift since we omit the land biosphere and sediments. Since quantification of the contribution of different mechanisms to the shift in mean marine $\delta^{13}C$ is still under debate, we limit ourselves to quantifying the potential effect of a mean shift in marine $\delta^{13}C$ on our analysis: A uniform marine shift of -0.2 ‰ and -0.4 ‰ is made to our LGM simulation to estimate the effect on our analysis of $\overline{BP_{eff}}$. In these cases, the best-fit LGM $\overline{BP_{eff}}$ lies closer to ~65 % and ~55 %, respectively (Fig. S19). We conclude that our analysis, which omits mean shifts in marine $\delta^{13}C$ due to sediment-weathering imbalances or land biosphere changes, provides an upper estimate of LGM $\overline{BP_{eff}}$. Future work could explore how changes in $\delta^{13}C^{atm}$ together with sedimentation-weathering imbalances could affect marine $\delta^{13}C$ in NorESM-OC. Much of the remaining model-proxy $\delta^{13}C$ data mismatch is due to a too weak vertical chemocline in glacial simulations. Therefore, scaling of the biological pump efficiency does not fully explain the glacial ocean proxy data using the modelled circulation field thought to most closely represent the glacial state. We see different explanations of the bias that could strengthen the glacial chemocline, such as the further aging (by isolation and stratification) of interior waters through improved simulation of deep-water formation, which would increase the regenerated signature of the interior LGM ocean. Especially Southern Source Waters would have a large potential (due to their large volume contribution) to increase global interior radiocarbon ages and regenerated signatures of the interior ocean.

The estimated $\overline{BP_{eff}}$ increase to 75 % corresponds to an increase in oceanic DIC ($\Delta DIC$) of ~1850 Gt C. This lies in the range of estimates as derived by the Bern3D Earth System Model of Intermediate Complexity constrained by single proxy targets ($pCO_2^{atm}$, $\delta^{13}C^{atm}$, marine $\delta^{13}C$ of DIC, or deep equatorial Pacific $CO_3^{2-}$). If all four targets are used as constraints, the range of $\Delta DIC$ estimates based on the Bern3D model is higher than 1850 Gt C (see

Fig. 7 and Appendix A). In order to disentangle and understand the processes contributing to ΔDIC, especially the large contribution from sedimentation-weathering imbalances, further work seems necessary.

Our results underline that only those coupled climate models that contain the processes and/or components that realistically change both ocean circulation and biogeochemistry will be able to simulate an LGM ocean in satisfactory agreement with proxy data. Such a simulation is also a test for Earth system models for their ability to reproduce natural climate variations adequately as a basis for reliable future projections, including human-induced forcing (e.g., Zhu et al., 2021). I.e., a satisfactory fidelity of Earth System Models in reproducing orbitally forced climate variations will increase our confidence in these models as tools for projecting future anthropogenic climate change. Therefore, future research should aim to identify the exact physical and biogeochemical processes that could have driven the increase in global mean biological pump efficiency (i.e., the interior regenerated signature) between the PI and LGM, with a likely central role for Southern Source Waters.

The general agreement between our model results and proxy data for ocean circulation (within the uncertainty of reconstructions), after adjustments to the sea surface salinity field, demonstrates an advantage of our forced ocean model setup, and its flexibility, over fully coupled Earth System Models in exploring different circulation and biological pump scenarios for explaining glacial $CO_2$ changes. We conclude that a large PI-to-LGM increase in the efficiency of the biological pump (from 38 to up to ~75 %) as well as a reorganization of ocean circulation/stratification are essential for simulating an LGM ocean in optimal agreement with proxy data.

## Appendix A

### 2.1 Experimental set-up of the Bern3D model

We estimate the LGM-PI change in marine DIC content (ΔDIC) by applying a reduced form emulator based on forcing-response relationships obtained from factorial simulation over the deglacial with the Bern3D model (see Jeltsch-Thömmes et al., 2019). The mean over 21 ka BP to 19 ka BP and over 500 to 200 yr BP are used as LGM and PI time intervals for the calculation of ΔDIC. To estimate ΔDIC, the forcing-response relationships of seven generic deglacial carbon cycle mechanisms in regard to LGM-PI changes in four observational targets ($p$CO$_2^{atm}$, $\delta^{13}$C$^{atm}$, marine $\delta^{13}$C of DIC, and deep equatorial Pacific $CO_3^{2-}$) and DIC are investigated with the Bern3D. The seven generic forcings comprise changes in wind stress, air-sea gas exchange, the export rain ratio between organic and inorganic carbon, the remineralization depth of organic particles, coral growth, weathering of organic material, and carbon stocks in the land biosphere. These seven generic deglacial carbon cycle mechanisms were varied individually by systematic parameter variations in addition to well-established forcings such as orbital parameters, greenhouse gas radiative forcing, land ice albedo, coral reef regrowth, and North Atlantic freshwater forcing. In a next step, Latin hypercube parameter sampling was used to vary the processes in combination and probe for nonlinear interactions and use the results to adjust the above forcing-response relationships. An emulator of the form

$$\Delta T(\Delta p_1, \dots, \Delta p_7) = a^T + b^T * (\sum_{i=1}^{7} \Delta p_i * S_i^T(\Delta p_i)),$$

is derived from the forcing response relationships. $S_i^T(\Delta p_i)$ is the sensitivity for each target $T$ to the change in each mechanism $\Delta p_i$. $a^T$ is the offset and $b^T$ the slope of the respective linear fit from the multi-parameter adjustment in order to correct for non-linearities in the addition of the seven generic mechanisms. We use the same half a

million parameter combinations as Jeltsch-Thömmes et al. (2019) and investigate ΔDIC. The four proxy targets ($p$CO$_2^{atm}$, $\delta^{13}$C$^{atm}$, marine $\delta^{13}$C of DIC, and deep equatorial Pacific CO$_3^{2-}$) are used as constraints.

## 2.2 Results from the Bern3D model

Applying single constraints only, yields similar ranges for ΔDIC for each constraint, covering the range of ΔDIC values from about 0 Gt C to 6000 Gt C (Fig. 7). Considering all four proxy targets simultaneously shifts the estimate of ΔDIC to higher values such that the covered range amounts to about 2400 to 5500 Gt C and a median estimate of about 3900 Gt C (±1σ range of [3350,4480] Gt C; Appenidx A, Fig. 7). This might seem large at first. Considering changes in the different carbon reservoirs that changed over the deglacial and eventually drew carbon from the ocean (atmosphere, land biosphere, coral reef regrowth, sedimentation-weathering imbalances) helps to put the estimated ΔDIC into context (see also Appendix A, Fig. 7). The atmospheric increase over the deglacial is well constrained from ice-cores and amounts to about 200 Gt C. The change in the land biosphere carbon storage is more uncertain. A recent estimate, which integrates multiple proxy-data, derives an increase of the land biosphere over the deglacial of about 850 Gt C (median estimate) (Jeltsch-Thömmes et al., 2019), in good accordance with another recent estimate based on a dynamic global vegetation model (Müller and Joos, 2020). The growth of coral reefs during sea level rise removed an additional 380 Gt C (Vecsei and Berger, 2004) to 1200 Gt C and more (Milliman, 1993; Kleypas, 1997; Ridgwell et al., 2003) from the ocean. Finally, imbalances in the sedimentation-weathering cycle of carbon need to be considered on the multi-millennial timescales. The

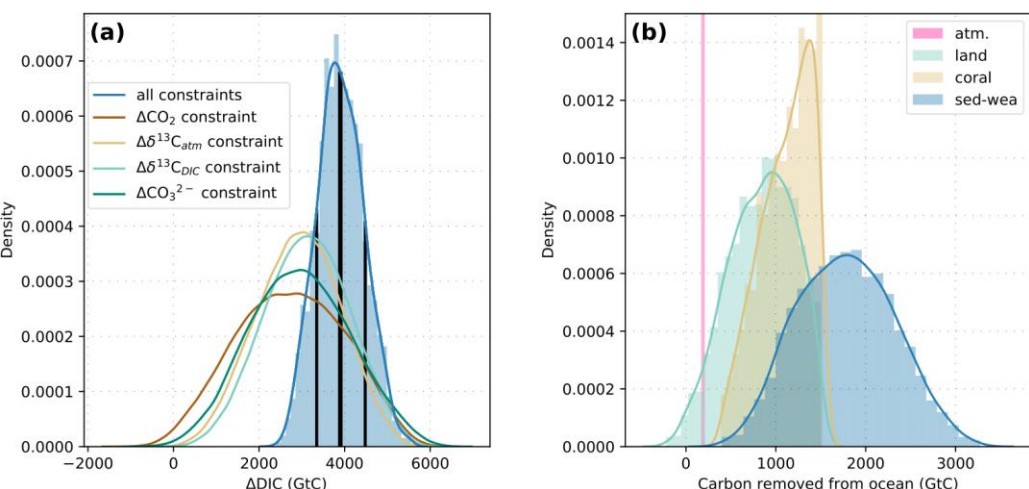

**Figure 7 (a) Histogram (blue bars, normalized) and kernel density estimate (blue line; all constraints) of ΔDIC (Δ is taken here as LGM minus PI) of parameter combinations fulfilling all four proxy constraints. Black vertical lines show the median ΔDIC estimate of ~3900 Gt C and the ±1σ range of [3350,4480] Gt C. The other four lines are kernel density estimates of ΔDIC for parameter combinations fulfilling only one of the four proxy constraints at a time. (b) Contributions to ΔDIC (all four proxy constraints fulfilled) from the atmosphere, land, corals, and sedimentation-weathering imbalances.**

contribution from sedimentation-weathering imbalances to ΔDIC is uncertain and covers a range from a removal of almost 0 Gt C to more than 3000 Gt C from the ocean as estimated with the Bern3D model. The results thus point to an important role and large contribution from sedimentation-weathering imbalances to ΔDIC estimates over glacial/interglacial timescales, however, with a large uncertainty. Finally, it has to be noted that in order to use the cost-efficient emulator and explore a large parameter space only the change between LGM and PI was considered. Including the spatio-temporal evolution of several proxies in transient model simulations will help to

further gain understanding of governing processes and narrow down the ΔDIC estimate, but is beyond the scope of this manuscript.

**Data availability.** The model output data for the last 30 years of the PI (Morée et al., 2020a) and LGM (Morée et al., 2020b) simulation are freely available at the NIRD Research Data Archive.

**Author contributions.** ALM, JS and CH conceptualized the study and developed the methodology in cooperation with IB and contribution from AJT. ALM did the formal analysis of the data and wrote the original draft for which she visualized the results. ALM, JS, CH, AJT and UN investigated the results. All authors contributed to the review and editing of the original draft. JS and ALM curated the data, contributed to the software and validated the results. CH and JS supervised the project, and provided the resources and funding acquisition needed for this work.

**Competing interests.** The authors declare that they have no conflict of interest.

**Disclaimer.** This article reflects only the authors' view – the funding agencies as well as their executive agencies are not responsible for any use that may be made of the information that the article contains.

**Acknowledgments.** This is a contribution to the Bjerknes Centre for Climate Research (Bergen, Norway). Storage and computing resources were provided by UNINETT Sigma2 - the National Infrastructure for High Performance
Computing and Data Storage in Norway (project numbers nn/ns2980k and nn/ns2345k). AM is grateful for PhD funding through the Faculty for Mathematics and Natural Sciences of the University of Bergen (UoB) as well as support by the Meltzer fund of UoB and Erasmus Mundus to stay at ETH Zürich for part of this work. JS acknowledges funding through the Research Council of Norway (project INES 270061). The NorESM components were upgraded with support from project EVA (Earth system modelling of climate Variations in the Anthropocene,
grant no. 229771, Research Council of Norway) and, CRESCENDO (Coordinated Research in Earth Systems and Climate: Experiments, kNowledge, Dissemination and Outreach; Horizon 2020 European Union's Framework Programme for Research and Innovation, grant no. 641816, European Commission). AJT acknowledges support from the Swiss National Science Foundation (# 200020_172476) and the Oeschger Centre for Climate Change Research.

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

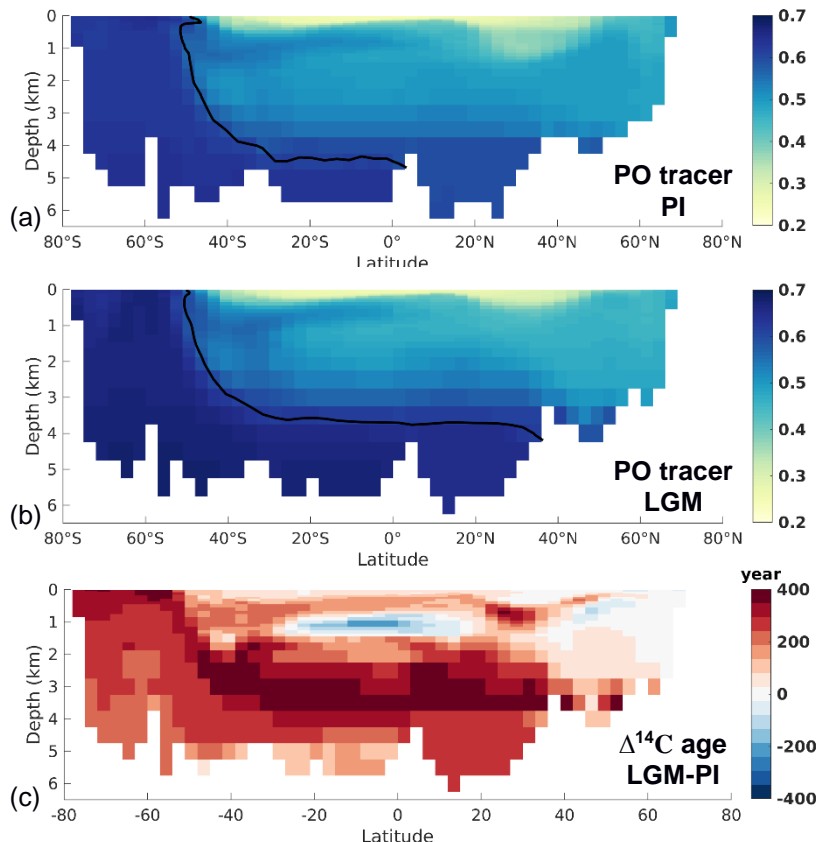

**Figure 1 Atlantic zonal mean of the conservative water mass tracer 'PO' (25-35° W) for (a) the PI and (b) the LGM model states. PO = $O_2$ + 172 * $PO_4$ (Broecker, 1974). The line represents the respective end-member PO of the Southern Source Waters (mean Southern Ocean surface PO), and thus the extent of Southern source water. (c) The change in radiocarbon age between the LGM and PI. See Fig. S11 for the corresponding Pacific zonal mean transects.**

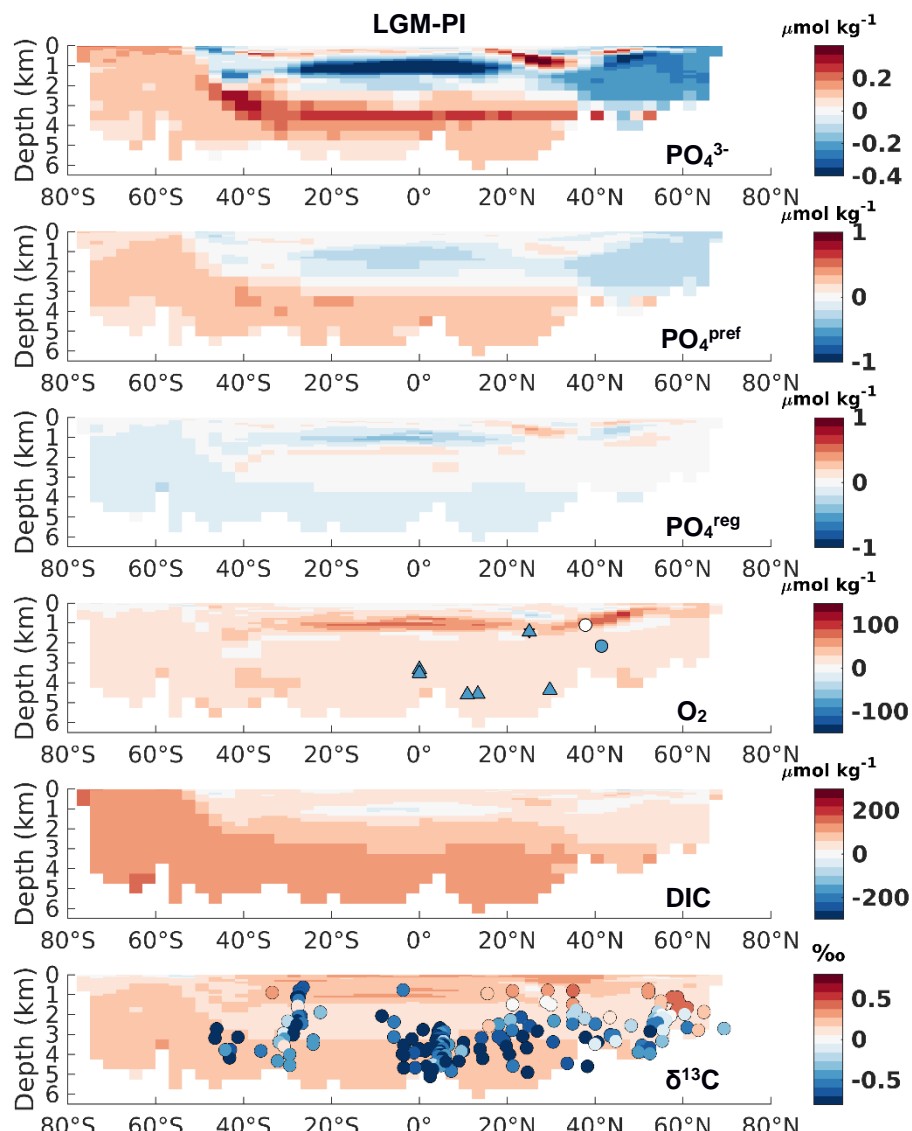

**Figure 2 Atlantic zonal mean (25-35° W) of LGM-PI changes for the original model output. See left-hand column of Fig. S17 for the corresponding Pacific sections. Overlay on O₂ is qualitative estimates of LGM-PI changes in O₂ within 30° from 30° W, with blue being a decrease, white indicating unclear changes and red indicating an increase in O₂ (Jaccard and Galbraith, 2011). Simulated LGM-PI δ13C is compared to a compilation of LGM minus Late Holocene δ13C data within 30° from 30° W (Peterson et al., 2014; Muglia et al., 2018; Molina-Kescher et al., 2016; Sikes et al., 2016; Burckel et al., 2016; Gottschalk et al., 2016b-c; Hodell and Channell, 2016; Howe and Piotrowski, 2017).**

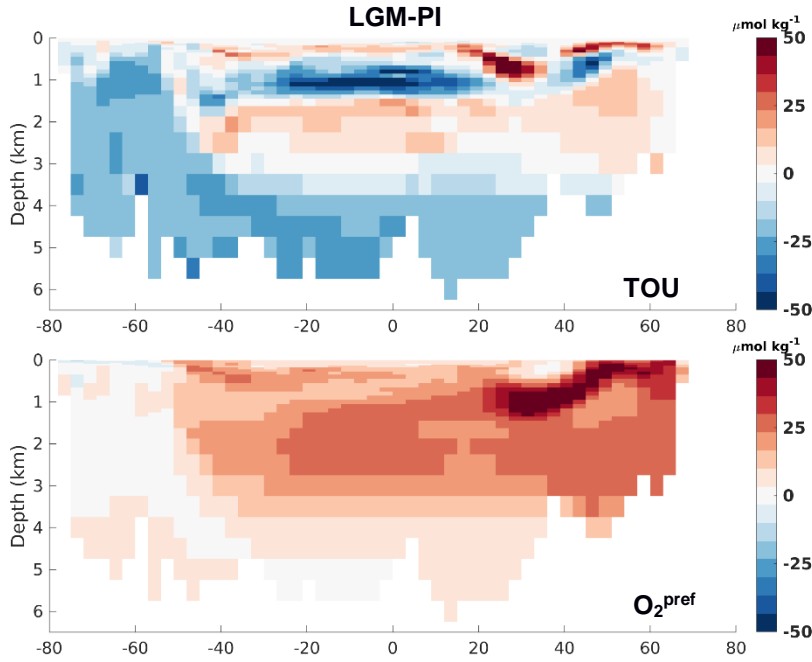

**Figure 3 Atlantic zonal mean (25-35° W) of simulated LGM-PI O₂ components True Oxygen Utilization (TOU= O₂^pref - O₂) (Ito et al., 2004) and preformed O₂ (O₂^pref), which is simulated as the O₂ concentration that leaves the surface ocean, and is thus different from saturated O₂ (Tjiputra et al., 2020).**

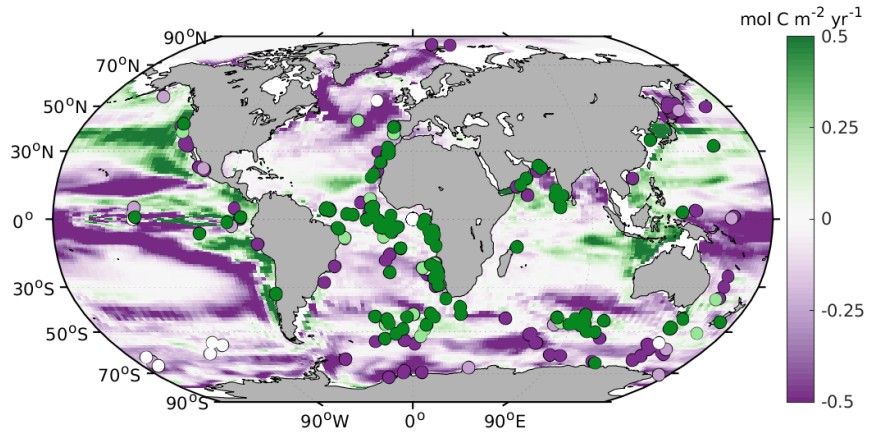

**Figure 4 Comparison between the simulated LGM-PI change in export production at 100m depth and Kohfeld et al. (2005) qualitative data (dots: dark purple=decrease, light purple=small decrease, white=no change, light green=light increase, dark green=increase).**

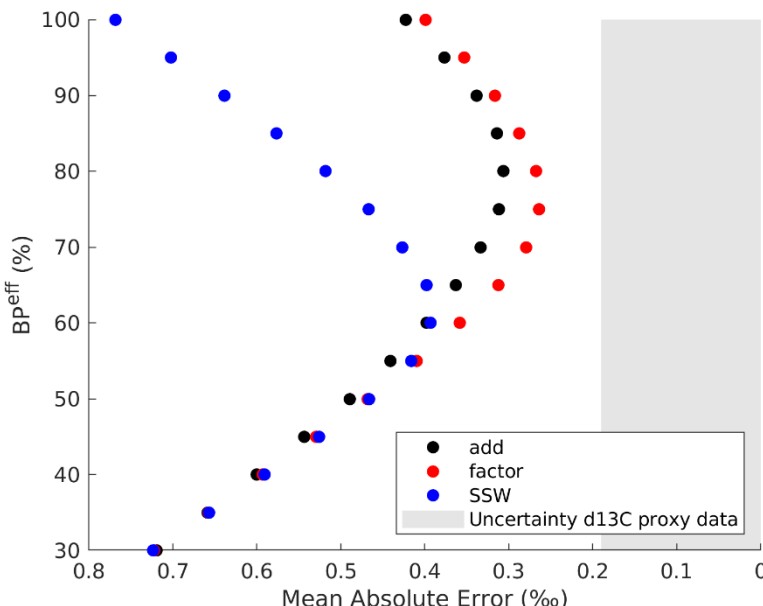

**Figure 5 Efficiency of the biological pump** ($\overline{BP_{eff}}$) **versus the mean absolute error between all δ¹³C proxy data and the nearest model grid-cell δ¹³C, for different methods (Sect. 2.4). Note that the original LGM** $\overline{BP_{eff}}$ **is 33 %. The δ¹³C sediment core data have an uncertainty of ~0.19‰, shaded in grey (Peterson et al., 2014).**

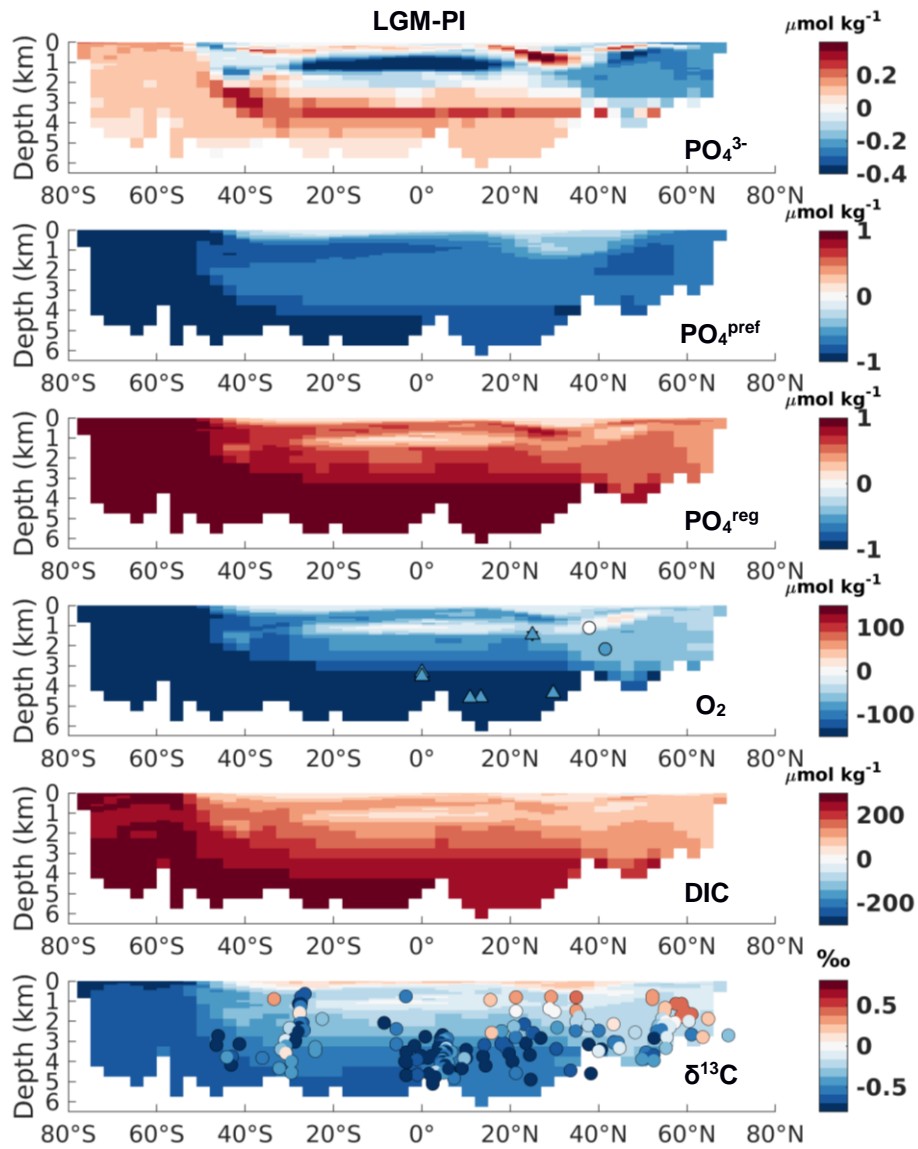

**Figure 6 As Figure 2, but for an adjusted $\overline{BP_{eff}}$ of 75 %. See right-hand column of Fig. S17 for the Pacific transects.**