# Peer review of "Evaluating the Biological Pump Efficiency of the Last Glacial Maximum Ocean using $\delta^{13}C$"

_Climate of the Past, 2020_

## Referee Comment (RC1) · Anonymous Referee #1 · 26 Mar 2020

Review of "Evaluating the Biological Pump Efficiency of the Last Glacial Maximum Ocean using d13C" by Moree et al.

The authors discussed about the glacial changes in d13C distribution in the ocean by comparing LGM ocean (NorESM-OC model) simulations with proxy data. The model significantly underestimates the glacial d13C changes compared with the proxy data; for example, negative signal of d13C in the deep Atlantic Ocean inferred from the proxy data is not reproduced in the model. At the same time, the model shows the decrease of the ocean biological pump efficiency in the LGM (33%) compared with the PI (38%), opposite to the fact that this is believed to be increased from the proxy data. The authors discussed the response of d13C by artificially increasing the ocean biological pump efficiency. The authors concluded that an approximate doubling of the global

mean biological pump efficiency from 38% (PI) to 75% (LGM) leads to the best-fit of d13C distribution between the model and the proxy. The manuscript deals with an important topic and contains interesting result which contributes to our understanding the glacial changes in the ocean carbon cycle. However, I think that the manuscript needs considerable revision. Followings are my comments about the manuscript, which I think needs to be seriously addressed before its publication.

Major comments

(1) The authors artificially increased the efficiency of the carbon pump at the LGM for their discussion. However, the mechanism behind this increase is not discussed enough in the manuscript. In other words, why do the original NorESM-OC model fail to simulate the glacial increase of the efficiency of the carbon pump? This needs to be more seriously discussed in the revised manuscript.

(2) Related to the above comment, the authors' conclusion "an approximate doubling of the global mean biological pump efficiency from 38% (PI) to 75% (LGM) reduces model-proxy biases the most" appears to depend highly on the reproducibility of their original LGM simulation. For example, the strength of the AMOC in the LGM simulation appears to significantly affect this number: the weaker AMOC tends to increase the efficiency whereas the stronger AMOC tends to decrease it. I request the authors to discuss about the robustness of their conclusion.

(3) I think that discussion about the effect on glacial changes in pCO2 is important. The authors stated that only 21 ppm lowering is found in their original LGM simulation. How much lowering of pCO2 is expected after the efficiency of the carbon pump is doubled in the LGM simulation?

Specific comments

Line15-26 (Abstract) : In my reading, I think that "relative roles of physical and biological changes" is not clearly evaluated in the manuscript.

Line23 (Abstract): The word "theoretical" appears not appropriate. ("potential" might be better)

Line26-35 (Abstract): I think that this sentence (which describes remaining issue and future work rather than the direct conclusion of the study) should be removed or shortened.

Section2.4: This is key section for understanding how the authors control the efficiency of the ocean carbon pump, but I feel that its description is not very clear and difficult to fully understand. For the demonstration, I request the authors to show the Figure of PO4_new after the adjustment by methods 1, 2, and 3, together with PO4_model.

Line28 (page 6): Definition of deltaPCO4(reg) is given at lines 1-4 on page7 but should be described before eqns. (2)-(3).

Line20-26 (page8): The discussion here is not clear for me. What do the authors mean by "the transition line in the PO tracer in Fig.1"?

Line2-28 (page11): The discussions made here are difficult to understand because the information on Bern3D is not given to readers at all.

Line16 (page11): What does deltaDIC stand for? Its definition is missing.

Line29-38 (page11): For the authors' reference, as for the discussion about O2, Yamamoto et al. (2019, Climate of the Past) discuss the role of glaciogenic dust in glacial O2 changes.

Line12-29 (page12): For the authors' reference, as for deep water formation processes in the Southern Ocean, Kobayashi et al. (2015, 2018; Paleoceangraphy) discuss about its representation in the OGCM and its potential role in glacial water mass age and ocean carbon cycle. This study appears closely related to the discussion the authors made here.

---

## Referee Comment (RC2) · Anonymous Referee #2 · 5 Apr 2020

General comments

The authors use an ocean-sea-ice model (NorESM-OC) that also includes biogeochemistry, $\delta$13C carbon isotopes and radiocarbon, to quantify the role of the efficiency of the LGM biological pump in obtain the best agreement between model simulations and proxy data. Their results indicate that the efficiency should be doubled to obtain the smallest model-proxy mismatch.

The model setup is novel, properly thought through, overall well-described, and can certainly be used to provide useful insight on long-standing questions about the role of ocean circulation and biogeochemistry in driving glacial-interglacial changes in ocean carbon storage. However, the structure and clarity of parts of the manuscript need to be substantially improved. Some additional simulations/sensitivity experiments may also

need to be included, or at least their potential implications need to be better discussed and compared with the existing literature. A few highly relevant studies, and all very recent, are also missing in the references.

Setting up these simulations must have involved a substantial amount of work and this should be acknowledged and this framework will also be useful to investigate other research questions. This study makes valuable contributions to the topic and definitely deserves to be published, but several issues need to be addressed first, as described in the comments below.

Specific comments

Abstract, page 2, line 33: This statement is a bit too strong. The LGM is indeed a good test case for models and their evaluation and process-based understanding, but it can't be considered a necessary "requirement" for their reliability for future projections. I get the point and I agree, but this need to be rephrased.

Page 2, lines 10 and 23: Add references to Stein et al. (2020) and Marzocchi and Jansen (2019), especially since these studies both address directly the role of physical changes on glacial carbon storage, which is not really done in this manuscript. These also needs to be discussed further with the results – see later comments.

Methods

The simulations are integrated for a long period of time. Nonetheless, it would still be useful to show some of the LGM ocean state equilibrium/drift in the Supplement. Perhaps some timeseries of T and S and/or AMOC and Drake Passage transport, which are already mentioned in the text.

The Bern3D model part of the study needs to be introduced and explained, at least briefly, in this section – with proper reference to the Supplement for the rest of the details.

Results and discussion

This part of the manuscript needs some substantial restructuring and improvements. Parts of it are quite confusing, which takes away from the key findings and the main points that the authors are trying to get across.

Perhaps separate more clearly parts of the results that are more of a "model evaluation" and then for each of these have a subsection that discuss the reasons for the biases, to give some separation between results and discussion, especially where comparisons to observations and other studies are also discussed.

All of this is already in the text, but currently quite mixed up all together, making several parts a little hard to follow. I am not against having results and discussion together, but the structure needs to be clearer and easier to follow.

Section 3.1 is a little hard to follow without any figures. . .maybe add some in the Supplement?

Discuss the radiocarbon ages also with respect to the results of Burke et al. (2015)

Line 31: add references to Jansen (2017) and Marzocchi and Jansen (2019) to support this statement on the importance of atmospheric temperatures for both LGM water masses and biogeochemistry, respectively.

Line 35: this needs to be discussed a little further (i.e. the underestimation of negative buoyancy fluxes) – for instance, compare Klockmann et al. (2018) – this is an example of where I think a separate Discussion section is missing. Alternatively, this could be picked up again in the conclusions as one of the potentially important biases.

The abyssal cell actually looks weaker at the LGM? (Figure S5) This also needs to be discussed, perhaps here.

Line 5: add reference to Marzocchi and Jansen (2019) and Stein et al. (2020) where

the link to ocean carbon storage is actually tested.

Section 3.2.2

Lines 10-26: This result (i.e. reduced LGM biological pump efficiency but lower pCO2 concentrations) is not dissimilar from what discussed in Marzocchi and Jansen (2019), despite a very different model setup. So this is worth discussing further – perhaps think about this in the context of the carbon pump decomposition. This may mean that there is something we simply don't understand in this part of the mechanism. Can your study clarify this apparent discrepancy further? Can you make this clearer/highlight it better?

Lines 10-19: this is another example where this is a discussion part, but it's somewhat "thrown" in the middle of some other text. So again this needs restructuring to make it easier for the reader to follow.

Line 25: here the reference is Marzocchi and Jansen (2019) rather than Jansen (2017).

Lines 2-21: This part about the Bern3D ESM comes a bit out of the blue and I can't say that this is explained well enough and entirely clear. Make better reference to the Supplement and better introduce the setup in the Methods (as noted before), where the goals of this additional step need to be better clarified and introduced. Then it will come less out of the blue here in the results.

Lines 12-29: this is again a somewhat self-standing discussion part that should perhaps be a subsection.

Here, and/or earlier, you should discuss the results of Odalen et al. (2019) Actually, would it be feasible to test their variable C/P ratio in your simulations?

Also could you quantify the dependence of your results to your model initial state, as discussed in Odalen et al. (2018)? [this reference is already cited in the manuscript].

Conclusions

Add a reference to Rae et al. (2019) when discussing the importance of southern-sourced waters. This should probably also be discussed earlier in the results/discussion.

Technical corrections

Abstract

Line 17: ocean model state? Do you mean "equilibrium simulations"? Clarify. Ocean model state is not the best term to use here.

Line 23: "we explore the theoretical effects" doesn't quite make sense. This could just be "we explore/test the effects".

Line 29: again "theoretical" is not quite the right word. Just say "our approach". Same in the rest of the manuscript (e.g. page 7, 10, 13). Perhaps do just call it "offline".

Page 10, line 30: miss-match should be mismatch.

Everywhere: "Southern Source" should really be "southern-sourced".

References

Burke, A., Stewart, A.L., Adkins, J.F., Ferrari, R., Jansen, M.F. and Thompson, A.F., 2015. The glacial mid‐depth radiocarbon bulge and its implications for the overturning circulation. Paleoceanography, 30(7), pp.1021-1039.

Klockmann, M., Mikolajewicz, U. and Marotzke, J., 2016. The effect of greenhouse gas concentrations and ice sheets on the glacial AMOC in a coupled climate model. Climate of the Past, 12, pp.1829-1846.

Marzocchi, A. and Jansen, M.F., 2019. Global cooling linked to increased glacial car-

bon storage via changes in Antarctic sea ice. Nature Geoscience, 12(12), pp.1001.

Ödalen, M., Nycander, J., Ridgwell, A., Oliver, K.I., Peterson, C.D. and Nilsson, J., 2019. Variable C/P composition of organic production and its effect on ocean carbon storage in glacial model simulations. Biogeosciences Discussions, pp.1-33. (accepted) DOI: https://doi.org/10.5194/bg-2019-149

Stein, K., Timmermann, A., Kwon, E.Y. and Friedrich, T., 2020. Timing and magnitude of Southern Ocean sea ice/carbon cycle feedbacks. Proceedings of the National Academy of Sciences, 117(9), pp.4498-4504.

---

## Author Comment (AC1) · 18 May 2020

Dear Referee #1,

Thank you for your time to provide constructive feedback on our manuscript 'Evaluating the Biological Pump Efficiency of the Last Glacial Maximum Ocean using δ13C'. A response to each of the comments is provided below (in italic text). Specifically, we propose to include a separate discussion section in a revised version of the manuscript. Here, the concerns of the reviewer on several discussion topics and missing references would be addressed. Additionally, we wish to improve the methods section by clarifying our approach to artificially enhance the efficiency of the biological pump (i.e., Sect. 2.4) and the use of the Bern3D model (new section 2.5).

Yours sincerely,
Anne Morée and co-authors

**Review of "Evaluating the Biological Pump Efficiency of the Last Glacial Maximum Ocean using d13C" by Moree et al.**

The authors discussed about the glacial changes in d13C distribution in the ocean by comparing LGM ocean (NorESM-OC model) simulations with proxy data. The model significantly underestimates the glacial d13C changes compared with the proxy data; for example, negative signal of d13C in the deep Atlantic Ocean inferred from the proxy data is not reproduced in the model. At the same time, the model shows the decrease of the ocean biological pump efficiency in the LGM (33%) compared with the PI (38%), opposite to the fact that this is believed to be increased from the proxy data. The authors discussed the response of d13C by artificially increasing the ocean biological pump efficiency. The authors concluded that an approximate doubling of the global mean biological pump efficiency from 38% (PI) to 75% (LGM) leads to the best-fit of d13C distribution between the model and the proxy. The manuscript deals with an important topic and contains interesting result which contributes to our understanding the glacial changes in the ocean carbon cycle. However, I think that the manuscript needs considerable revision. Followings are my comments about the manuscript, which I think needs to be seriously addressed before its publication.

**Major comments**

(1) The authors artificially increased the efficiency of the carbon pump at the LGM for their discussion. However, the mechanism behind this increase is not discussed enough in the manuscript. In other words, why do the original NorESM-OC model fail to simulate the glacial increase of the efficiency of the carbon pump? This needs to be more seriously discussed in the revised manuscript.

*Author response: We will revise the manuscript as outlined below.*

*Changes in the manuscript: We will address this comment in two ways. First, we will revise section 2.4 to clarify how we artificially increased the efficiency of the biological pump (see also our reply to the comment on Sect 2.4). Secondly, we will extend our discussion by including a new discussion section at the end of the paper. Here, a more detailed and structured discussion on the lack of a simulated increase in the biological pump efficiency will be given. Specifically, we will discuss both physical (e.g., stratification, solubility pump, isolation and strength of abyssal overturning cell) and biogeochemical (e.g., export production, remineralization rate) mechanisms that could contribute to an increased efficiency of the biological pump - and whether NorESM-OC is able to capture these. We want to stress however that identification of the exact mechanisms is beyond the scope of our manuscript. Earth System Models are generally found to incompletely capture the biogeochemistry and strengthening of the biological pump for the LGM ocean, and identification of the exact processes that are missing in these models is a major challenge in modelling the LGM ocean (e.g., Galbraith and Skinner, 2020).*

(2) Related to the above comment, the authors' conclusion "an approximate doubling of the global mean biological pump efficiency from 38% (PI) to 75% (LGM) reduces model-proxy biases the most" appears to depend highly on the reproducibility of their original LGM simulation. For example, the strength of the AMOC in the LGM simulation appears to significantly affect this number: the weaker AMOC tends to increase the efficiency whereas the stronger AMOC tends to decrease it. I request the authors to discuss about the robustness of their conclusion.

*Author response: We plan to insert the following explanation and discussion into the discussion and conclusion at the end of our manuscript.*

*Changes in the manuscript: Changes between preindustrial and LGM ocean circulation fields as simulated by ocean models generally fail to account for the 100-120 ppm drawdown in atmospheric CO2 (taken the outgassing by the land biosphere into account) when used in global ocean carbon cycle models (Heinze et al., 1991; Brovkin et al., 2007). The induced change is usually too small. Correspondingly, also the vertical d13C gradient (Deltadelta13C) is often not fully reproduced to its full extent. If we assume that the simulated circulation changes are realistic, this indicates that one needs to employ additional biogeochemical or ecological processes to enhance the atmospheric CO2 drawdown by the ocean and to enhance the biological pump. This can be done either by artificially enhancing the pump efficiency (which we explore in our theoretical framework) or by changing the nutrient cycling, e.g. by adjusting the stoichiometric ratio of elements N:P:C away from the Redfield ratio values or by adding nutrients to the ocean. Changing the pump efficiency is an easy way to implement the effect needed, leaving open the exact process that leads to this effect. A more sluggish ocean circulation, already leads to a partial increase in pump efficiency, because smaller amounts of nutrients are brought to the ocean surface and get exported in a more slowly overturning ocean, while the particle flux still operates with unchanged gravity acceleration. This leads to partial carbon and nutrient fractionation between upper and deep ocean, but not enough to explain the full CO2 reduction as observed in the atmosphere.*

(3) I think that discussion about the effect on glacial changes in pCO2 is important. The authors stated that only 21 ppm lowering is found in their original LGM simulation. How much lowering of pCO2 is expected after the efficiency of the carbon pump is doubled in the LGM simulation?

*Author response: The additional carbon inventory in the ocean corresponding to a doubling of the efficiency of the biological pump is quantified at ~1850 Gt C (p. 11, l.16). Where this additional carbon would have come from (the land, ocean sediments or atmosphere) is something we can not distinguish in our model setup or our offline exploration of the potential effects of changes in the efficiency of the biological pump. Nevertheless, the magnitude of this estimated change in marine DIC (i.e., ~1850 Gt C) allows for full (~80 ppm more than simulated, which is ~ 170 Gt C) draw-down to LGM atmospheric carbon concentrations, a profound decrease in land carbon (which could be ~850 Gt C as estimated by Jeltsch-Thömmes et al., 2019) as well as a source of DIC from the deep ocean sediments/CaCO3. We see it would be of interest to discuss this in the manuscript, and will include this in a revised version.*

*Changes in the manuscript: Extension of the discussion to include information on the potential effects of a doubling of the efficiency of the biological pump on atmospheric pCO2.*

**Specific comments**

Line15-26 (Abstract): In my reading, I think that "relative roles of physical and biological changes" is not clearly evaluated in the manuscript.

*Author response: This sentence is meant to describe that we explored the net effect of physical changes (f.e., circulation, temperature, atmospheric forcing, land-sea mask) and biogeochemical changes (different dust field, offline exploration of the potential effects of an increased efficiency of the biological pump) in shaping the LGM ocean (and specifically its δ13C distribution). As we do not present a range of different physical ocean states, we see that rephrasing of this sentence is appropriate. Related to this, we would rephrase p.2 l. 12-13*

*and p.12 l. 31-33 to clarify that we simulated LGM-PI changes in both the physical and biogeochemical state of the ocean and study its cumulative effect on d13C.*

*Changes in the manuscript: Revise sentence 'This modelling study explores the relative roles of physical and biological changes in the ocean needed to simulate an LGM ocean in satisfactory agreement with proxy data, and here especially δ13C.' to 'This modelling study presents a realization of the physical and biological changes in the ocean needed to simulate an LGM ocean in satisfactory agreement with proxy data, and here especially δ13C.' Additionally, revise p.2 l. 12-13 and p.12 l. 31-33 to clarify that we simulated LGM-PI changes in both the physical and biogeochemical state of the ocean and study its cumulative effect on δ13C.*

Line23 (Abstract): The word "theoretical" appears not appropriate. ("potential" might be better)

*Author response: We think that 'potential (offline)' would best summarize that we explored the potential effects of different efficiencies of the biological pump without doing additional modelling experiments. Similarly we would revise the other occurrences of the word 'theoretical' to clarify we mean exploring the potential (and offline) effects when we describe our approach.*

*Changes in the manuscript: Replace 'theoretical' with 'potential (offline)' on p.1 l.23. Additionally, rephrase other occurrences of the word theoretical throughout the text to clarify our intention to explore the potential (offline) effects whenever we describe our approach.*

Line26-35 (Abstract): I think that this sentence (which describes remaining issue and future work rather than the direct conclusion of the study) should be removed or shortened.

*Author response: As the model-proxy data mismatch is one of the central results of the study, we do wish to include this in the abstract. Nevertheless, the discussion of the reasons for this mismatch could indeed be shortened, and we will do so in a revised version of our manuscript.*

*Changes in the manuscript: Shorten p.1 l. 28-35.*

Section2.4: This is key section for understanding how the authors control the efficiency of the ocean carbon pump, but I feel that its description is not very clear and difficult to fully understand. For the demonstration, I request the authors to show the Figure of PO4_new after the adjustment by methods 1, 2, and 3, together with PO4_model.

*Author response: Thank you for making us aware that the different methods of distributing additional regenerated PO4 are not entirely clear in the current version of the manuscript. We will be able to include a demonstration figure (in the SM, for the Atlantic) as requested, which shows how the 3 different methods of adding regenerated PO4 will alter the regenerated PO4 distribution (for one biological pump efficiency) relative to the model regenerated PO4 distribution. In addition, we will update p.6 l.35 to p.7 l.5 to improve the clarity of the text.*

*Changes in the manuscript: We will add a figure to the SM to visualize the differences between the 3 different methods and clarify the explanation of the methods (p.6 l.35 to p.7 l.5).*

Line28 (page 6): Definition of deltaPCO4(reg) is given at lines 1-4 on page7 but should be described before eqns. (2)-(3).

*Author response: Lines 1-4 on p.7 describe how the total global change in deltaPO4(reg) is distributed over the grid for the 3 different methods, while p.6 l.28 defines deltaPO4(reg) for a specific grid-cell which is relevant for the updated fields of O2, DIC and d13C. We understand the current description is confusing, and will therefore clarify the explanation of the methods and definitions (p.6 l.28 to p.7 l.5) in the text.*

*Changes in the manuscript: We will clarify the explanation of the methods and definitions (p.6 l.28 to p.7 l.5) in the text.*

Line20-26 (page8): The discussion here is not clear for me. What do the authors mean by "the transition line in the PO tracer in Fig.1"?

*Author response: We note that the line in Fig. 1, which is the mean SSW PO value, is too thin. Besides that, we see that a more thorough introduction of the PO tracer and how it was used here will help the reader to understand Fig. 1.*
*Changes in the manuscript: We will thicken the line in Fig. 1 and extend the caption of Fig. 1 as well as the text in section 3.2.1 (l.20-26) to clarify our use and interpretation of the PO tracer.*

Line2-28 (page11): The discussions made here are difficult to understand because the information on Bern3D is not given to readers at all.
*Author response: The Bern3D model is mentioned in SM3 and in Sect. 3.3, and we see there is a need for a clearer introduction of the Bern3D model in the main text and how it was used in our study (see also our reply to the next comment), and we will address this by adding a new a new subsection under Methods.*
*Changes in the manuscript: We will add a new subsection 2.5 to describe the purpose and technical details of the Bern3D model and how it is used to estimate ΔDIC.*

Line16 (page11): What does deltaDIC stand for? Its definition is missing.
*Author response: ΔDIC is defined at its first occurrence on p. 11, l. 3 as 'the LGM-PI change in marine DIC'. Here, LGM for ΔDIC is the mean over 21 kyr BP to 19 kyr BP and PI is the mean of 500 to 200 yr BP. We see that this definition, together with the technical information on the Bern3D model (In the SM 3 and Sect. 3.3) could be lifted to a new subsection (Sect. 2.5) under Methods for clarity, which also addresses the previous comment.*
*Changes in the manuscript: We will add a new subsection 2.5 to describe the purpose and technical details of the Bern3D model and how it is used to estimate ΔDIC.*

Line29-38 (page11): For the authors' reference, as for the discussion about O2, Yamamoto et al. (2019, Climate of the Past) discuss the role of glaciogenic dust in glacial O2 changes.
*Author response: Thank you for making us aware of this interesting paper. We will include its results in our discussion on O2. This paper also highlights the importance of using a glacial dust field when looking at the biogeochemistry of the LGM ocean. As changing the dust field in the LGM simulation is the only change to the model which can directly affect the biogeochemical model through relief of iron limitation, we think it is worth it to also include the reference in our methods section (p.5 l.24) to explain the interest of using the Lambert et al. (2015) dust dataset to force our model.*
*Changes in the manuscript: Include the results of Yamamoto et al. (2019) in our discussion on the LGM-PI O2 changes as well as to argue for the use of a glacial dust field in our methods section.*

Line12-29 (page12): For the authors' reference, as for deep water formation processes in the Southern Ocean, Kobayashi et al. (2015, 2018; Paleoceangraphy) discuss about its representation in the OGCM and its potential role in glacial water mass age and ocean carbon cycle. This study appears closely related to the discussion the authors made here.
*Author response: Thank you for making us aware of these Kobayashi et al. studies from 2015 and 2018. We agree that including their findings in our discussion would improve this part of the manuscript, and we will do so in a revised version.*
*Changes in the manuscript: Include the findings of Kobayashi et al. (2015; 2018) in our discussion on the remaining mode-proxy data mismatch.*

*References of the response*
*Brovkin, V., Ganopolski, A., Archer, D., and Rahmstorf, S.: Lowering of glacial atmospheric CO2 in response to changes in oceanic circulation and marine biogeochemistry, Paleoceanography, 22, 10.1029/2006PA001380, 2007.*
*Galbraith, E. D., and Skinner, L. C.: The Biological Pump During the Last Glacial Maximum, Annual Review of Marine Science, 12, 559-586, 10.1146/annurev-marine-010419-010906, 2020.*

Heinze, C., Maier-Reimer, E., and Winn, K.: Glacial pCO2 Reduction by the World Ocean: Experiments With the Hamburg Carbon Cycle Model, Paleoceanography, 6, 395-430, 10.1029/91PA00489, 1991.

Jeltsch-Thömmes, A., Battaglia, G., Cartapanis, O., Jaccard, S. L., and Joos, F.: Low terrestrial carbon storage at the Last Glacial Maximum: constraints from multi-proxy data, Climate of the Past, 15, 849-879, 10.5194/cp-15-849-2019, 2019.

Kobayashi, H., A. Abe-Ouchi, and A. Oka: Role of SouthernOcean stratification in glacial atmospheric CO2 reduction evaluated by a three-dimensional ocean general circulation model, Paleoceanography, 30, 1202–1216, 10.1002/2015PA002786, 2015.

Kobayashi, H., & Oka, A.: Response of atmospheric pCO2 to glacial changes in the Southern Ocean amplified by carbonate compensation, Paleoceanography and Paleoclimatology, 33, 1206–1229, 10.1029/2018PA003360, 2018.

Yamamoto, A., Abe-Ouchi, A., Ohgaito, R., Ito, A., and Oka, A.: Glacial CO2 decrease and deep-water deoxygenation by iron fertilization from glaciogenic dust, Clim. Past, 15, 981–996, 10.5194/cp-15-981-2019, 2019.

---

## Author Comment (AC2) · 18 May 2020

Dear Referee #2,

Thank you for your time and effort to provide constructive feedback on our manuscript. We have replied to each of your comments and concerns (in italic text) below. Specifically, we will include a separate discussion section in a revised version of our manuscript. In this section, the missing references that the reviewer pointed out will be included. We expect this change will address the reviewers' concerns about the clarity and structure of the text, as well as improve the discussion of potential implications of our results and the comparison with existing literature.

Yours sincerely,
Anne Morée and co-authors

**General comments**
The authors use an ocean-sea-ice model (NorESM-OC) that also includes biogeochemistry, δ13C carbon isotopes and radiocarbon, to quantify the role of the efficiency of the LGM biological pump in obtain the best agreement between model simulations and proxy data. Their results indicate that the efficiency should be doubled to obtain the smallest model-proxy mismatch.
The model setup is novel, properly thought through, overall well-described, and can certainly be used to provide useful insight on long-standing questions about the role of ocean circulation and biogeochemistry in driving glacial-interglacial changes in ocean carbon storage. However, the structure and clarity of parts of the manuscript need to be substantially improved. Some additional simulations/sensitivity experiments may also need to be included, or at least their potential implications need to be better discussed and compared with the existing literature. A few highly relevant studies, and all very recent, are also missing in the references.
Setting up these simulations must have involved a substantial amount of work and this should be acknowledged and this framework will also be useful to investigate other research questions. This study makes valuable contributions to the topic and definitely deserves to be published, but several issues need to be addressed first, as described in the comments below.

**Specific comments**
Abstract, page 2, line 33: This statement is a bit too strong. The LGM is indeed a good test case for models and their evaluation and process-based understanding, but it can't be considered a necessary "requirement" for their reliability for future projections. I get the point and I agree, but this need to be rephrased.
*Author response: We think the reviewer refers to p. 1 l. 33 here. We will change the statement as well as shorten this part of the abstract (see also our response to reviewer (#1, response to comment on Line 26-35). Our results underline that only those coupled climate models that contain the processes and/or components that realistically change both ocean circulation and biogeochemistry will be able to simulate an LGM ocean in satisfactory agreement with proxy data. Such a simulation is also a test for Earth system models for their ability to reproduce natural climate variations adequately as a basis for reliable future projections, including human-induced forcing. I.e., a satisfactory fidelity of Earth System Models in reproducing orbitally forced climate variations will increase our confidence in these models as tools for projecting future anthropogenic climate change.*
*Changes in the manuscript: Shorten abstract lines 26-35 and move part of text to discussion/conclusion sections.*

Page 2, lines 10 and 23: Add references to Stein et al. (2020) and Marzocchi and Jansen (2019), especially since these studies both address directly the role of physical changes on glacial carbon storage, which is not really done in this manuscript. These also needs to be discussed further with the results – see later comments.
*Author response: Agreed.*

*Changes in the manuscript: The references Stein et al. (2020) and Marzocchi and Jansen (2019) will be added and discussed.*

Methods
The simulations are integrated for a long period of time. Nonetheless, it would still be useful to show some of the LGM ocean state equilibrium/drift in the Supplement. Perhaps some timeseries of T and S and/or AMOC and Drake Passage transport, which are already mentioned in the text.
The Bern3D model part of the study needs to be introduced and explained, at least briefly, in this section – with proper reference to the Supplement for the rest of the details.
*Author response:*
*We will include a time series over the last 1000 years of the LGM and PI simulations in the supplement (for S, T, AMOC and Drake Passage transport) to give a visual impression of the equilibration/drift. Regarding the Bern3D model, we see that besides the information on the Bern3D model in Sect. 3.3 and SM3, the model and its application in the context of our study should be introduced in the methods section as well for which we will include a new section.*
*Changes in the manuscript: Addition of a new equilibration time series figure in the supplement, as well as a new methods Section (2.5) to describe the setup and use of the Bern3D model in this study.*

Results and discussion
This part of the manuscript needs some substantial restructuring and improvements. Parts of it are quite confusing, which takes away from the key findings and the main points that the authors are trying to get across.
Perhaps separate more clearly parts of the results that are more of a "model evaluation" and then for each of these have a subsection that discuss the reasons for the biases, to give some separation between results and discussion, especially where comparisons to observations and other studies are also discussed.
All of this is already in the text, but currently quite mixed up all together, making several parts a little hard to follow. I am not against having results and discussion together, but the structure needs to be clearer and easier to follow.
*Author response: In order to improve the clarity and structure of section 3, we can agree that the inclusion of a separate discussion section would help. We would be able to lift some of the model-data comparison discussion points that are currently spread throughout the Sect. 3 text into such a new section, as well as provide a dedicated section for the discussion of the remaining model-data mismatch after adjustment of the efficiency of the biological pump.*
*Changes in the manuscript: Inclusion of a separate discussion section at the end of the manuscript that focuses on the discussion around model-proxy data mismatches. This section can consist of two parts, one discussing the model-data mismatch of the original simulation, and a second dedicated to the remaining model-data mismatch after adjustment of the efficiency of the biological pump (i.e., p. 12 l. 12-29).*

Section 3.1 is a little hard to follow without any figures. . .maybe add some in the Supplement?
*Author response: We see that no reference is made in Sect. 3.1 to Fig. S5, which shows the PI physical state and could already be referred to here (currently done in Sect 3.2). The biogeochemical state of the PI simulation is described in detail for the C isotopes in Tjiputra et al. (2020). Otherwise, our focus is on the change in the biogeochemical marine state (LGM-PI), which is shown in Fig. 2. To address the reviewers comment further, we can provide supplementary figures of PI temperature (section), PP (vertically integrated), and regenerated phosphate (section) compared to observational estimates, since these are mainly discussed in Sect. 3.1.*
*Changes in the manuscript: Include reference to Fig. S5 in Sect 3.1 as well as new supplementary figures of temperature, PP and regenerated PO4 compared to observations.*

*Page 8*
Discuss the radiocarbon ages also with respect to the results of Burke et al. (2015)
*Author response: We will include the Burke et al. (2015) reference in our discussion.*
*Changes in the manuscript: Add and discuss Burke et al. (2015).*

Line 31: add references to Jansen (2017) and Marzocchi and Jansen (2019) to support this statement on the importance of atmospheric temperatures for both LGM water masses and biogeochemistry, respectively.
*Author response: We agree these references should be cited here and will do so in a revised version of the manuscript.*
*Changes in the manuscript: Add Jansen (2017) and Marzocchi and Jansen (2019) references.*

Line 35: this needs to be discussed a little further (i.e. the underestimation of negative buoyancy fluxes) – for instance, compare Klockmann et al. (2018) – this is an example of where I think a separate Discussion section is missing. Alternatively, this could be picked up again in the conclusions as one of the potentially important biases. The abyssal cell actually looks weaker at the LGM? (Figure S5) This also needs to be discussed, perhaps here.
*Author response: As described in our response to the general comment on the result and discussions section, we propose to include a separate discussion section in a revised version of the manuscript. In the first part of this new section, where we want to discuss the model-proxy data mismatch of the original simulation we would be able to discuss our results in more detail regarding buoyancy fluxes and the abyssal cell strength (which indeed weakens). We assume the reviewer means Klockmann et al. (2016) here (as in their reference list), and will include the findings of Klockmann et al. (2016) in our discussion.*
*Changes in the manuscript: Discuss our simulation with regard to (Southern Ocean) buoyancy fluxes and the strength of the abyssal cell in the new discussion section. Specifically, include and discuss Klockmann et al. (2016).*

*Page 9*
Line 5: add reference to Marzocchi and Jansen (2019) and Stein et al. (2020) where the link to ocean carbon storage is actually tested.
*Author response: See our response to p. 2, l 10 and 23 above.*
*Changes in the manuscript: We will include these references here and in other appropriate locations where their findings are useful for our discussion.*

*Section 3.2.2*
Lines 10-26: This result (i.e. reduced LGM biological pump efficiency but lower $pCO_2$ concentrations) is not dissimilar from what discussed in Marzocchi and Jansen (2019), despite a very different model setup. So this is worth discussing further – perhaps think about this in the context of the carbon pump decomposition. This may mean that there is something we simply don't understand in this part of the mechanism. Can your study clarify this apparent discrepancy further? Can you make this clearer/highlight it better?
*Author response: The lowered atmospheric $pCO_2$ is expected from the combined effect of the ocean volume decrease and increased $CO_2$ solubility due to decreased ocean temperatures (p. 9 l. 19-22). That is, mostly the physical C pump is represented in our study and driving down atmospheric $pCO_2$ (as evidenced by increases in $DIC_{pref}$ and $DIC_{sat}$, not shown (for definitions see also Sect 3.1 in Tjiputra et al., (2020) and references therein)). The lack (and actually decreased efficiency) of a soft tissue pump strengthening is discussed in Sect. 3.3. The inability of our model to simulate the strengthening of the soft tissue pump is expected from earlier results for ESMs and our model setup (f.e. summary point 5 in Galbraith and Skinner, 2020; p. 9 l. 22-26) - and indeed indicates that some biogeochemical processes/mechanisms are lacking in these models. Pinning down the exact processes of this strengthening is an ongoing challenge, and beyond the scope of our study. Nevertheless, we would be able to decompose the LGM-PI change in DIC into $DIC_{soft}$, $DIC_{pref}$, $DIC_{sat}$, $DIC_{bio}$,*

*DICcarb and DICdiss (definitions in Sect 3.1 in Tjiputra et al., (2020) and references therein) and add a figure of this to the supplement in order to visualize their individual contributions. We will highlight this result and its discussion in the new discussion section.*
*Changes in the manuscript: Clarify the atmospheric pCO2 drawdown in the context of the LGM-PI changes in the different C pump components (DICsoft, DICpref, DICsat, DICbio, DICcarb and DICdiss) in a new supplement figure. Discuss this and specifically the lack of a contribution from the soft tissue pump on simulated LGM atmospheric pCO2 in more detail in the new discussion section.*

*Page 10*
Lines 10-19: this is another example where this is a discussion part, but it's somewhat "thrown" in the middle of some other text. So again this needs restructuring to make it easier for the reader to follow.
*Author response: As described in our response to the general comment on the result and discussions section, we propose to include a separate discussion section in a revised version of the manuscript. The discussion on p. 10 l. 10-19 could be moved to such a new section to improve the structure of Sect. 3.*
*Changes in the manuscript: Include p. 10 l. 10-19 in a new discussion section.*

Line 25: here the reference is Marzocchi and Jansen (2019) rather than Jansen (2017).
*Author response: Thank you for noting this, we see that Marzocchi and Jansen (2019) is more appropriate here than Jansen (2017) and will adjust the manuscript accordingly*
*Changes in the manuscript: As suggested*

*Page 11*
Lines 2-21: This part about the Bern3D ESM comes a bit out of the blue and I can't say that this is explained well enough and entirely clear. Make better reference to the Supplement and better introduce the setup in the Methods (as noted before), where the goals of this additional step need to be better clarified and introduced. Then it will come less out of the blue here in the results.
*Author response: As described above, we will add a new methods section 2.5 on the Bern3D model. Here we will pay specific attention to clarifying why and how the Bern3D model was used.*
*Changes in the manuscript: Addition of a new methods section 2.5 on the Bern3D model.*

*Page 12*
Lines 12-29: this is again a somewhat self-standing discussion part that should perhaps be a subsection.
*Author response: As described in our response to the general comment on the result and discussions section, we propose to include a separate discussion section in a revised version of the manuscript. The second part of this new section will be dedicated to discussing the remaining model-proxy data error after adjustment of the efficiency of the biological pump, which is essentially p. 12 l. 12-29.*
*Changes in the manuscript: Restructure the text to include a new discussion section, where p. 12 l. 12-29 would be a second section that discusses the remaining model-proxy data error after adjustment of the efficiency of the biological pump.*

Here, and/or earlier, you should discuss the results of Odalen et al. (2019). Actually, would it be feasible to test their variable C/P ratio in your simulations?
*Author response: We assume that the reviewer refers to the paper Ödalen et al. (2020). Their results, which could decrease in d13C while keeping (regenerated) PO4 constant, could indeed be included in our discussion on the remaining proxy-data mismatch. With regard to the feasibility to test variable C:P ratios in our model setup: Our model is computationally more*

*demanding than the cGENIE model as employed by Ödalen et al (2020) and we do not have the resources for repeating long runs for this currently.*
*Changes in the manuscript: Discuss the results on variations in the C:P ratio by Ödalen et al. (2020) and their potential implications for our remaining model-data mismatch.*

Also could you quantify the dependence of your results to your model initial state, as discussed in Odalen et al. (2018)? [this reference is already cited in the manuscript].
*Author response:*
*We did not carry out experiments with vastly different initial states. It is known since long, that different initial conditions for temperature and salinity can result in different circulation modes. However, in our case we assume that the initial conditions for the glacial ocean circulation would not be too different from preindustrial conditions and not fully different. Due to the high computational demand of our model, we cannot carry out multiple spin-ups (as 10,000 years done in the cGENIE model) with different initial conditions or tunings. This would not be feasible given currently available computational resources.*
*Changes in the manuscript: No changes will be made.*

*Conclusions*
Add a reference to Rae et al. (2019) when discussing the importance of southern-sourced waters. This should probably also be discussed earlier in the results/discussion.
*Author response: We assume that the reviewer refers to the paper Rae et al. (2018). As our paper does not deal with pH changes, we do not specifically discuss this article, but can include this reference as it highlights the central role of SSW as the reviewer points out.*
*Changes in the manuscript: Rae et al. (2018) will be cited at page 2, lines 5 and 23, and added to the reference list.*

*Technical corrections*
Abstract Line 17: ocean model state? Do you mean "equilibrium simulations"? Clarify. Ocean model state is not the best term to use here.
*Author response: We will adjust the text as proposed below*
*Changes in the manuscript: replace sentence 'We prepared a PI and LGM ocean model state (NorESM-OC) with full biogeochemistry (including the carbon isotopes δ13C and radiocarbon) and dynamic sea ice.' with 'We prepared a PI and LGM equilibrium simulation using model NorESM-OC with full biogeochemistry (including the carbon isotopes δ13C and radiocarbon) and dynamic sea ice.'*

Line 23: "we explore the theoretical effects" doesn't quite make sense. This could just be "we explore/test the effects".
*Author response: We think clarifying that our approach is exploring the potential effects only (i.e. it is an approximation as no actual simulation is done) is important here, and we therefore propose to replace 'theoretical' with 'potential (offline)' in the abstract.*
*Changes in the manuscript: Replace 'theoretical' with 'potential (offline)' on p.1 l.23.*

Line 29: again "theoretical" is not quite the right word. Just say "our approach". Same in the rest of the manuscript (e.g. page 7, 10, 13). Perhaps do just call it "offline".
*Author response: See our response to the previous comment.*
*Changes in the manuscript: Replace 'theoretical' with 'potential (offline)' or 'our approach' throughout the text to clarify our intention to explore the potential (offline) effects whenever we describe our approach.*

Page 10, line 30: miss-match should be mismatch.
*Author response: Thank you for noting this mistake, we will adjust the manuscript as suggested.*
*Changes in the manuscript: change p. 10, l. 30 miss-match to mismatch.*

Everywhere: "Southern Source" should really be "southern-sourced".
*Author response: We revisited the literature and see that both southern source water (e.g., Adkins, 2013; Curry and Oppo, 2005; Roberts et al., 2010) and southern-sourced water (Howe et al., 2016; Pöppelmeier et al., 2018) are commonly used. We therefore feel the current use of Southern Source Water (SSW) throughout the manuscript can be maintained.*
*Changes in the manuscript: None.*

*References*
Burke, A., Stewart, A.L., Adkins, J.F., Ferrari, R., Jansen, M.F. and Thompson, A.F., 2015. The glacial middepth radiocarbon bulge and its implications for the overturning circulation. Paleoceanography, 30(7), pp.1021-1039.
Klockmann, M., Mikolajewicz, U. and Marotzke, J., 2016. The effect of greenhouse gas concentrations and ice sheets on the glacial AMOC in a coupled climate model. Climate of the Past, 12, pp.1829-1846.
Marzocchi, A. and Jansen, M.F., 2019. Global cooling linked to increased glacial carbon storage via changes in Antarctic sea ice. Nature Geoscience, 12(12), pp.1001.
Ödalen, M., Nycander, J., Ridgwell, A., Oliver, K.I., Peterson, C.D. and Nilsson, J., 2019. Variable C/P composition of organic production and its effect on ocean carbon storage in glacial model simulations. Biogeosciences Discussions, pp.1-33. (accepted) DOI: https://doi.org/10.5194/bg-2019-149
Stein, K., Timmermann, A., Kwon, E.Y. and Friedrich, T., 2020. Timing and magnitude of Southern Ocean sea ice/carbon cycle feedbacks. Proceedings of the National Academy of Sciences, 117(9), pp.4498-4504

*References of the response*
*Adkins, J. F.: The role of deep ocean circulation in setting glacial climates, Paleoceanography, 28, 539-561, 10.1002/palo.20046, 2013.*
*Burke, A., Stewart, A. L., Adkins, J. F., Ferrari, R., Jansen, M. F., and Thompson, A. F.: The glacial mid-depth radiocarbon bulge and its implications for the overturning circulation, Paleoceanography, 30, 1021-1039, 10.1002/2015PA002778, 2015.*
*Curry, W. B., and Oppo, D. W.: Glacial water mass geometry and the distribution of δ13C of ΣCO2 in the western Atlantic Ocean, Paleoceanography, 20, 10.1029/2004PA001021, 2005.*
*Galbraith, E. D., and Skinner, L. C.: The Biological Pump During the Last Glacial Maximum, Annual Review of Marine Science, 12, 559-586, 10.1146/annurev-marine-010419-010906, 2020.*
*Klockmann, M., Mikolajewicz, U., and Marotzke, J.: The effect of greenhouse gas concentrations and ice sheets on the glacial AMOC in a coupled climate model, Clim. Past, 12, 1829-1846, 10.5194/cp-12-1829-2016, 2016.*
*Roberts, N. L., Piotrowski, A. M., McManus, J. F., and Keigwin, L. D.: Synchronous Deglacial Overturning and Water Mass Source Changes, Science, 327, 75, 10.1126/science.1178068, 2010.*
*Howe, J. N. W., Piotrowski, A. M., Noble, T. L., Mulitza, S., Chiessi, C. M., and Bayon, G.: North Atlantic Deep Water Production during the Last Glacial Maximum, Nature Communications, 7, 11765, 10.1038/ncomms11765, 2016.*
*Jansen, M. F.: Glacial ocean circulation and stratification explained by reduced atmospheric temperature, Proceedings of the National Academy of Sciences, 114, 45-50, 10.1073/pnas.1610438113, 2017.*
*Marzocchi, A., and Jansen, M. F.: Global cooling linked to increased glacial carbon storage via changes in Antarctic sea ice, Nature Geoscience, 12, 1001-1005, 10.1038/s41561-019-0466-8, 2019.*
*Pöppelmeier, F., Gutjahr, M., Blaser, P., Keigwin, L. D., and Lippold, J.: Origin of Abyssal NW Atlantic Water Masses Since the Last Glacial Maximum, Paleoceanography and Paleoclimatology, 33, 530-543, 10.1029/2017PA003290, 2018.*

Rae, J. W. B., Burke, A., Robinson, L. F., Adkins, J. F., Chen, T., Cole, C., Greenop, R., Li, T., Littley, E. F. M., Nita, D. C., Stewart, J. A., and Taylor, B. J.: CO2 storage and release in the deep Southern Ocean on millennial to centennial timescales, Nature, 562, 569-573, 10.1038/s41586-018-0614-0, 2018.

Stein, K., Timmermann, A., Kwon, E. Y., and Friedrich, T.: Timing and magnitude of Southern Ocean sea ice/carbon cycle feedbacks, Proceedings of the National Academy of Sciences, 117, 4498, 10.1073/pnas.1908670117, 2020.

Ödalen, M., Nycander, J., Ridgwell, A., Oliver, K. I. C., Peterson, C. D., and Nilsson, J.: Variable C/P composition of organic production and its effect on ocean carbon storage in glacial-like model simulations, Biogeosciences, 17, 2219–2244, https://doi.org/10.5194/bg-17-2219-2020, 2020.

---

## Author Response (AR1)

Dear Referee #1,

Thank you for your time to provide constructive feedback on our manuscript 'Evaluating the Biological Pump Efficiency of the Last Glacial Maximum Ocean using $\delta^{13}C$'. A response to each of the comments is provided below (in italic text). Specifically, we include a separate discussion section in a revised version of the manuscript. Here, the concerns of the reviewer on several discussion topics and missing references are addressed. Additionally, we improved the methods section by clarifying our approach to artificially enhance the efficiency of the biological pump (i.e., Sect. 2.4) and the use of the Bern3D model (new section 2.5).

Yours sincerely,

Anne Morée and co-authors

**Major comments**

(1) The authors artificially increased the efficiency of the carbon pump at the LGM for their discussion. However, the mechanism behind this increase is not discussed enough in the manuscript. In other words, why do the original NorESM-OC model fail to simulate the glacial increase of the efficiency of the carbon pump? This needs to be more seriously discussed in the revised manuscript.

*Author response: We revised the manuscript as outlined below.*

*Changes in the manuscript: We addressed this comment in two ways. First, we revised section 2.4 to clarify how we artificially increased the efficiency of the biological pump (see also our reply to the comment on Sect 2.4). Secondly, we extended our discussion by including a new discussion section at the end of the paper (Sect. 4). Here, a more detailed and structured discussion on the lack of a simulated increase in the biological pump efficiency is given. Specifically, we discuss both physical (e.g., stratification, solubility pump, isolation and strength of abyssal overturning cell) and biogeochemical (e.g., export production, remineralization rate) mechanisms that could contribute to an increased efficiency of the biological pump - and whether NorESM-OC is able to capture these. We want to stress however that identification of the exact mechanisms is beyond the scope of our manuscript. Earth System Models are generally found to incompletely capture the biogeochemistry and strengthening of the biological pump for the LGM ocean, and identification of the exact processes that are missing in these models is a major challenge in modelling the LGM ocean (e.g., Galbraith and Skinner, 2020).*

(2) Related to the above comment, the authors' conclusion "an approximate doubling of the global mean biological pump efficiency from 38% (PI) to 75% (LGM) reduces model-proxy biases the most" appears to depend highly on the reproducibility of their original LGM simulation. For

example, the strength of the AMOC in the LGM simulation appears to significantly affect this number: the weaker AMOC tends to increase the efficiency whereas the stronger AMOC tends to decrease it. I request the authors to discuss about the robustness of their conclusion.

*Author response: Changes between preindustrial and LGM ocean circulation fields as simulated by ocean models generally fail to account for the 100-120 ppm drawdown in atmospheric pCO₂ (taken the outgassing by the land biosphere into account) when used in global ocean carbon cycle models (Heinze et al., 1991; Brovkin et al., 2007). The induced change is usually too small. Correspondingly, also the vertical d13C gradient (Deltadelta13C) is often not fully reproduced to its full extent. If we assume that the simulated circulation changes are realistic, this indicates that one needs to employ additional biogeochemical or ecological processes to enhance the atmospheric pCO₂ drawdown by the ocean and to enhance the biological pump. This can be done either by artificially enhancing the pump efficiency (which we explore in our theoretical framework) or by changing the nutrient cycling, e.g. by adjusting the stoichiometric ratio of elements N:P:C away from the Redfield ratio values or by adding nutrients to the ocean. Changing the pump efficiency is an easy way to implement the effect needed, leaving open the exact process that leads to this effect. A more sluggish ocean circulation, already leads to a partial increase in pump efficiency, because smaller amounts of nutrients are brought to the ocean surface and get exported in a more slowly overturning ocean, while the particle flux still operates with unchanged gravity acceleration. This leads to partial carbon and nutrient fractionation between upper and deep ocean, but not enough to explain the full pCO₂ reduction as observed in the atmosphere.*

*Changes in the manuscript: We included the above discussion in the new discussion section 4.*

(3) I think that discussion about the effect on glacial changes in pCO2 is important. The authors stated that only 21 ppm lowering is found in their original LGM simulation. How much lowering of pCO2 is expected after the efficiency of the carbon pump is doubled in the LGM simulation?

*Author response: The additional carbon inventory in the ocean corresponding to a doubling of the efficiency of the biological pump is quantified at ~1850 Gt C (p. 11, l.16). Where this additional carbon would have come from (the land, ocean sediments or atmosphere) is something we can not distinguish in our model setup or our offline exploration of the potential effects of changes in the efficiency of the biological pump. Nevertheless, the magnitude of this estimated change in marine DIC (i.e., ~1850 Gt C) allows for full (~80 ppm more than simulated, which is ~ 170 Gt C) draw-down to LGM atmospheric carbon concentrations, a profound decrease in land carbon (which could be ~850 Gt C as estimated by Jeltsch-Thömmes et al., 2019) as well as a source of DIC from the deep ocean sediments/CaCO₃. We see it would be of interest to discuss this in the manuscript, and will include this in a revised version.*

*Changes in the manuscript: Extension of the discussion to include information on the potential effects of a doubling of the efficiency of the biological pump on atmospheric pCO2 as discussed above.*

**Specific comments**

Line15-26 (Abstract): In my reading, I think that "relative roles of physical and biological changes" is not clearly evaluated in the manuscript.

*Author response: This sentence is meant to describe that we explored the net effect of physical changes (e.g., circulation, temperature, atmospheric forcing, land-sea mask) and biogeochemical changes (different dust field, offline exploration of the potential effects of an increased efficiency of the biological pump) in shaping the LGM ocean (and specifically its $\delta^{13}C$ distribution). As we do not present a range of different physical ocean states, we see that rephrasing of this sentence is appropriate. Related to this, we would rephrase p.2 l. 12-13 and p.12 l. 31-33 to clarify that we simulated LGM-PI changes in both the physical and biogeochemical state of the ocean and study its cumulative effect on $\delta^{13}C$.*

*Changes in the manuscript: We revised sentence 'This modelling study explores the relative roles of physical and biological changes in the ocean needed to simulate an LGM ocean in satisfactory agreement with proxy data, and here especially $\delta^{13}C$.' to 'This modelling study presents a realization of the physical and biological changes in the ocean needed to simulate an LGM ocean in satisfactory agreement with proxy data, and here especially $\delta^{13}C$.' Additionally, we revised p.2 l. 12-13 and p.12 l. 31-33 to clarify that we simulated LGM-PI changes in both the physical and biogeochemical state of the ocean and study its cumulative effect on $\delta^{13}C$.*

Line23 (Abstract): The word "theoretical" appears not appropriate. ("potential" might be better)

*Author response: We think that 'potential (offline)' would best summarize that we explored the potential effects of different efficiencies of the biological pump without doing additional modelling experiments. Similarly we would revise the other occurrences of the word 'theoretical' to clarify we mean exploring the potential (and offline) effects when we describe our approach.*

*Changes in the manuscript: We clarified the use of the word 'theoretical' throughout the text by adding the word '(offline)' or 'potential' to clarify our intention to explore the potential (offline) effects whenever we describe our approach.*

Line26-35 (Abstract): I think that this sentence (which describes remaining issue and future work rather than the direct conclusion of the study) should be removed or shortened.

*Author response: As the model-proxy data mismatch is one of the central results of the study, we do wish to mention this in the abstract, but move the detailed discussion out of the abstract.*

*Changes in the manuscript: The discussion of the reasons for the model- proxy data mismatch is moved to the new discussion section 4.*

Section2.4: This is key section for understanding how the authors control the efficiency of the ocean carbon pump, but I feel that its description is not very clear and difficult to fully understand.

For the demonstration, I request the authors to show the Figure of PO4_new after the adjustment by methods 1, 2, and 3, together with PO4_model.

*Author response: Thank you for making us aware that the different methods of distributing additional regenerated PO$_4$ are not entirely clear in the current version of the manuscript. We were able to include a demonstration figure as requested (Fig S4, for the Atlantic and an increase in BP_eff to 75%), which shows how the 3 different methods of adding regenerated PO$_4$ will alter the regenerated PO$_4$ distribution. In addition, we updated p.6 l.35 to p.7 l.5 to improve the clarity of this section of text. Note that the total PO$_4$ concentration is kept constant (only redistributions between regenerated and preformed PO$_4$ are considered).*

*Changes in the manuscript: We added a new figure to the SM to visualize the differences between the 3 different methods for regenerated PO$_4$, and clarify the explanation of the methods in the main text (p.6 l.35 to p.7 l.5).*

Line28 (page 6): Definition of deltaPCO4(reg) is given at lines 1-4 on page7 but should be described before eqns. (2)-(3).

*Author response: Lines 1-4 on p.7 describe how the total global change in deltaPO4(reg) is distributed over the grid for the 3 different methods, while p.6 l.28 defines deltaPO4(reg) for a specific grid-cell which is relevant for the updated fields of O2, DIC and d13C. We understand the current description is confusing, and therefore clarified the explanation of the methods and definitions (p.6 l.28 to p.7 l.5) in the text (see also our response to the previous comment).*

*Changes in the manuscript: We clarified the explanation of the methods and definitions (p.6 l.28 to p.7 l.5) in the text.*

Line20-26 (page8): The discussion here is not clear for me. What do the authors mean by "the transition line in the PO tracer in Fig.1"?

*Author response: We note that the line in Fig. 1, which is the mean SSW PO value, is too thin. Besides that, we see that a more thorough introduction of the PO tracer and how it was used here will help the reader to understand Fig. 1.*

*Changes in the manuscript: We thickened the transition line in Fig. 1 and extended the caption of Fig. 1 as well as the text in section 3.2.1 (l.20-26) to clarify our use and interpretation of the PO tracer. We also thickened the line in the corresponding Pacific figures in Fig. S12.*

Line2-28 (page11): The discussions made here are difficult to understand because the information on Bern3D is not given to readers at all.

*Author response: The Bern3D model is mentioned in SM3 and in Sect. 3.3, and we see there is a need for a clearer introduction of the Bern3D model in the main text and how it was used in our*

*study (see also our reply to the next comment), and we addressed this by adding a new subsection under Methods.*

*Changes in the manuscript: We added a new subsection 2.5 to describe the purpose and technical details of the Bern3D model and how it is used to estimate ΔDIC.*

Line16 (page11): What does deltaDIC stand for? Its definition is missing.

*Author response: ΔDIC is defined at its first occurrence on p. 11, l. 3 as 'the LGM-PI change in marine DIC'. Here, LGM for ΔDIC is the mean over 21 kyr BP to 19 kyr BP and PI is the mean of 500 to 200 yr BP. We see that this definition, together with the technical information on the Bern3D model (In the SM 3 and Sect. 3.3) could be lifted to a new subsection (Sect. 2.5) under Methods for clarity, which also addresses the previous comment.*

*Changes in the manuscript: We added a new subsection 2.5 as well as Appendix A to describe the purpose and technical details of the Bern3D model and how it is used to estimate ΔDIC.*

Line29-38 (page11): For the authors' reference, as for the discussion about O2, Yamamoto et al. (2019, Climate of the Past) discuss the role of glaciogenic dust in glacial O2 changes.

*Author response: Thank you for making us aware of this interesting paper. We included its results in our discussion on O2. This paper also highlights the importance of using a glacial dust field when looking at the biogeochemistry of the LGM ocean. As changing the dust field in the LGM simulation is the only change to the model which can directly affect the biogeochemical model through relief of iron limitation, we included the reference in our methods section as well (original p.5 l.24) to explain the interest of using the Lambert et al. (2015) dust dataset to force our model.*

*Changes in the manuscript: We included the results of Yamamoto et al. (2019) in our discussion on the LGM-PI $O_2$ changes as well as to argue for the use of a glacial dust field in our methods section.*

Line12-29 (page12): For the authors' reference, as for deep water formation processes in the Southern Ocean, Kobayashi et al. (2015, 2018; Paleoceanography) discuss about its representation in the OGCM and its potential role in glacial water mass age and ocean carbon cycle. This study appears closely related to the discussion the authors made here.

*Author response: Thank you for making us aware of these Kobayashi et al. studies from 2015 and 2018. We agree that including their findings in our discussion would improve this part of the manuscript, and we have done so in the revised version of our manuscript. Specifically, we mention now in Sect. 3.2.1 our simulation of the slowdown of the abyssal overturning cell as well as salinification of SSW while referring to the results of Kobayashi et al. (2015). We also mention the Kobayashi et al. (2015) study when underlining that both physical and biological changes must have occurred between the LGM and PI oceans. Note that we are not able to compare our*

*study directly to the study by Kobayashi and Oka (2018) as we excluded our sediment model and riverine fluxes in our simulations due to computational costs.*

*Changes in the manuscript: We included the findings of Kobayashi et al. (2015; 2018) in our discussion on the remaining model-proxy data mismatch. The references are added to the reference list.*

*References of the response*

*Brovkin, V., Ganopolski, A., Archer, D., and Rahmstorf, S.: Lowering of glacial atmospheric CO2 in response to changes in oceanic circulation and marine biogeochemistry, Paleoceanography, 22, 10.1029/2006PA001380, 2007.*

*Galbraith, E. D., and Skinner, L. C.: The Biological Pump During the Last Glacial Maximum, Annual Review of Marine Science, 12, 559-586, 10.1146/annurev-marine-010419-010906, 2020.*

*Heinze, C., Maier-Reimer, E., and Winn, K.: Glacial pCO2 Reduction by the World Ocean: Experiments With the Hamburg Carbon Cycle Model, Paleoceanography, 6, 395-430, 10.1029/91PA00489, 1991.*

*Jeltsch-Thömmes, A., Battaglia, G., Cartapanis, O., Jaccard, S. L., and Joos, F.: Low terrestrial carbon storage at the Last Glacial Maximum: constraints from multi-proxy data, Climate of the Past, 15, 849-879, 10.5194/cp-15-849-2019, 2019.*

*Kobayashi, H., A. Abe-Ouchi, and A. Oka: Role of SouthernOcean stratification in glacial atmospheric CO2 reduction evaluated by a three-dimensional ocean general circulation model, Paleoceanography, 30, 1202–1216, 10.1002/2015PA002786, 2015.*

*Kobayashi, H., & Oka, A.: Response of atmospheric pCO2 to glacial changes in the Southern Ocean amplified by carbonate compensation, Paleoceanography and Paleoclimatology, 33, 1206–1229, 10.1029/2018PA003360, 2018.*

*Yamamoto, A., Abe-Ouchi, A., Ohgaito, R., Ito, A., and Oka, A.: Glacial CO2 decrease and deep-water deoxygenation by iron fertilization from glaciogenic dust, Clim. Past, 15, 981–996, 10.5194/cp-15-981-2019, 2019.*

Dear Referee #2,

Thank you for your time and effort to provide constructive feedback on our manuscript. We have replied to each of your comments and concerns below (in italic text). Specifically, we included a separate discussion section in the revised version of our manuscript. In this section, the missing references that the reviewer pointed out are included. We hope the reviewer agrees with us that we addressed the reviewers' concerns about the clarity and structure of the text, as well as improved the discussion of potential implications of our results and the comparison with existing literature.

Yours sincerely,

Anne Morée and co-authors

**Specific comments**

Abstract, page 2, line 33: This statement is a bit too strong. The LGM is indeed a good test case for models and their evaluation and process-based understanding, but it can't be considered a necessary "requirement" for their reliability for future projections. I get the point and I agree, but this need to be rephrased.

*Author response: We think the reviewer refers to p. 1 l. 33 here. We will change the statement as well as shorten this part of the abstract (see also our response to reviewer (#1, response to comment on Line 26-35). Our results underline that only those coupled climate models that contain the processes and/or components that realistically change both ocean circulation and biogeochemistry will be able to simulate an LGM ocean in satisfactory agreement with proxy data. Such a simulation is also a test for Earth system models for their ability to reproduce natural climate variations adequately as a basis for reliable future projections, including human-induced forcing. I.e., a satisfactory fidelity of Earth System Models in reproducing orbitally forced climate variations will increase our confidence in these models as tools for projecting future anthropogenic climate change.*

*Changes in the manuscript: We shortened abstract lines 26-35 and moved part of text to the new discussion section.*

Page 2, lines 10 and 23: Add references to Stein et al. (2020) and Marzocchi and Jansen (2019), especially since these studies both address directly the role of physical changes on glacial carbon storage, which is not really done in this manuscript. These also needs to be discussed further with the results – see later comments.

*Author response: Agreed. We added the references to the original p2, lines 10&23 and extended our discussion on the physical changes in the new discussion Sect. 4.*

*Changes in the manuscript: The references Stein et al. (2020) and Marzocchi and Jansen (2019) are added to original p2, lines 10&23 and discussed in the new discussion Sect . 4. The references are added to the reference list.*

**Methods**

The simulations are integrated for a long period of time. Nonetheless, it would still be useful to show some of the LGM ocean state equilibrium/drift in the Supplement. Perhaps some timeseries of T and S and/or AMOC and Drake Passage transport, which are already mentioned in the text.

The Bern3D model part of the study needs to be introduced and explained, at least briefly, in this section – with proper reference to the Supplement for the rest of the details.

*Author response: We included a time series over the last 1000 years of the LGM and PI simulations in the supplement (for S, T, AMOC and Drake Passage transport, Fig. S5 and S6) to give a visual impression of the equilibration/drift. Referral to this new figure is made in the beginning of Sect. 3 as well as in 3.2.1 where the LGM physical ocean state is presented. Regarding the Bern3D model, we see that besides the information on the Bern3D model in Sect. 3.3 and SM3, the model and its application in the context of our study could be introduced in the methods section as well, for which we included a new section and appendix.*

*Changes in the manuscript: Addition of a new equilibration time series figure in the supplement and referral to this in the main text, as well as a new methods Section (2.5) and Appendix A to describe the setup and use of the Bern3D model in this study.*

**Results and discussion**

This part of the manuscript needs some substantial restructuring and improvements. Parts of it are quite confusing, which takes away from the key findings and the main points that the authors are trying to get across.

Perhaps separate more clearly parts of the results that are more of a "model evaluation" and then for each of these have a subsection that discuss the reasons for the biases, to give some separation between results and discussion, especially where comparisons to observations and other studies are also discussed.

All of this is already in the text, but currently quite mixed up all together, making several parts a little hard to follow. I am not against having results and discussion together, but the structure needs to be clearer and easier to follow.

*Author response: In order to improve the clarity and structure of section 3, we can agree that the inclusion of a separate discussion section would help and include this in the revised version of our manuscript. We were able to lift some of the model-data comparison discussion points that are currently spread throughout the Sect. 3 text into the new section, as well as provide a*

*dedicated section for the discussion of the remaining model-data mismatch after adjustment of the efficiency of the biological pump.*

*Changes in the manuscript: We included a separate discussion section at the end of the manuscript that focuses on the discussion around model-proxy data mismatches. This section consists of two parts, one discussing the model-data mismatch of the original simulation, and a second dedicated to the remaining model-data mismatch after adjustment of the efficiency of the biological pump (i.e., p. 12 l. 12-29).*

Section 3.1 is a little hard to follow without any figures. . .maybe add some in the Supplement?

*Author response: We see that no reference is made in Sect. 3.1 to Fig. S5, which shows the PI physical state and could already be referred to here (currently done in Sect 3.2). The biogeochemical state of the PI simulation is described in detail for the C isotopes in Tjiputra et al. (2020). Otherwise, our focus is on the change in the biogeochemical marine state (LGM-PI), which is shown in Fig. 2. To address the reviewer's comment further, we now provide supplementary figures of PI temperature (section), PP (vertically integrated), and regenerated phosphate (section) compared to observational estimates, since these are mainly discussed in Sect. 3.1.*

*Changes in the manuscript: We included reference to the original Fig. S5 in Sect 3.1 as well as new supplementary figures of temperature, PP and regenerated PO4 compared to observations.*

**Page 8**

Discuss the radiocarbon ages also with respect to the results of Burke et al. (2015)

*Author response: We now include the Burke et al. (2015) reference in our discussion.*

*Changes in the manuscript: We added and discussed Burke et al. (2015) when describing our simulation in Sect. 3.2.1.*

Line 31: add references to Jansen (2017) and Marzocchi and Jansen (2019) to support this statement on the importance of atmospheric temperatures for both LGM water masses and biogeochemistry, respectively.

*Author response: We agree these references should be cited here and did so in the revised version of the manuscript.*

*Changes in the manuscript: We added the Jansen (2017) and Marzocchi and Jansen (2019) references.*

Line 35: this needs to be discussed a little further (i.e. the underestimation of negative buoyancy fluxes) – for instance, compare Klockmann et al. (2018) – this is an example of where I think a separate Discussion section is missing. Alternatively, this could be picked up again in the

conclusions as one of the potentially important biases. The abyssal cell actually looks weaker at the LGM? (Figure S5) This also needs to be discussed, perhaps here.

*Author response: As described in our response to the general comment on the result and discussions section, we now include a separate discussion section in the revised version of the manuscript. In the first part of this new section, where we discuss the model-proxy data mismatch of the original simulation we discuss among other things our results in more detail regarding buoyancy fluxes and the abyssal cell strength (which indeed weakens). We assume the reviewer means Klockmann et al. (2016) here (as in their reference list) and included the findings of Klockmann et al. (2016) in this discussion.*

*Changes in the manuscript: We now discuss our simulation with regard to (Southern Ocean) buoyancy fluxes and the strength of the abyssal cell in the new discussion section. Specifically, we also included and discussed Klockmann et al. (2016).*

**Page 9**

Line 5: add reference to Marzocchi and Jansen (2019) and Stein et al. (2020) where the link to ocean carbon storage is actually tested.

*Author response: See our response to p. 2, l 10 and 23 above.*

*Changes in the manuscript: We included these references here and in other appropriate locations where their findings are useful for our discussion. The references are added to the reference list.*

*Section 3.2.2* Lines 10-26: This result (i.e. reduced LGM biological pump efficiency but lower pCO2 concentrations) is not dissimilar from what discussed in Marzocchi and Jansen (2019), despite a very different model setup. So this is worth discussing further – perhaps think about this in the context of the carbon pump decomposition. This may mean that there is something we simply don't understand in this part of the mechanism. Can your study clarify this apparent discrepancy further? Can you make this clearer/highlight it better?

*Author response: The lowered atmospheric $pCO_2$ is expected from the combined effect of the ocean volume decrease and increased $CO_2$ solubility due to decreased ocean temperatures (p. 9 l. 19-22). That is, mostly the physical C pump is represented in our study and driving down atmospheric $pCO_2$ (as evidenced by increases in DICpref and DICsat, see new discussion section). The lack (and actually decreased efficiency) of a soft tissue pump strengthening is also discussed in the original Sect. 3.3. The inability of our model to simulate the strengthening of the soft tissue pump is expected from earlier results for ESMs and our model setup (e.g. summary point 5 in Galbraith and Skinner, 2020; p. 9 l. 22-26) - and indeed indicates that some biogeochemical processes/mechanisms are lacking in these models. Pinning down the exact processes of this strengthening is an ongoing challenge, and beyond the scope of our study. Nevertheless, we decomposed the LGM-PI change in DIC into DICsoft, DICpref, DICsat, DICbio, DICcarb and DICdiss (Bernardello et al., 2014) and added a figure of this to the supplement in*

*order to visualize their individual contributions. We highlight this result and its discussion in the new discussion section.*

*Changes in the manuscript: We clarify the atmospheric $pCO_2$ drawdown despite the decrease in BPeff in the context of the LGM-PI changes in the different C pump components (DICsoft, DICpref, DICsat, DICbio, DICcarb and DICdiss) in the new discussion section 4, and included in a new supplement figure Fig. S17.*

**Page 10**

Lines 10-19: this is another example where this is a discussion part, but it's somewhat "thrown" in the middle of some other text. So again this needs restructuring to make it easier for the reader to follow.

*Author response: As described in our response to the general comment on the result and discussions section, we included a separate discussion section in the revised version of the manuscript. The discussion on p. 10 l. 10-19 is moved to the new discussion section to improve the structure of Sect. 3.*

*Changes in the manuscript: We included p. 10 l. 10-19 in the new discussion section.*

Line 25: here the reference is Marzocchi and Jansen (2019) rather than Jansen (2017).

*Author response: Thank you for noting this, we see that Marzocchi and Jansen (2019) is more appropriate here than Jansen (2017) and will adjust the manuscript accordingly*

*Changes in the manuscript: As suggested*

**Page 11**

Lines 2-21: This part about the Bern3D ESM comes a bit out of the blue and I can't say that this is explained well enough and entirely clear. Make better reference to the Supplement and better introduce the setup in the Methods (as noted before), where the goals of this additional step need to be better clarified and introduced. Then it will come less out of the blue here in the results.

*Author response: As described above, we added a new methods section 2.5 and Appendix A on the Bern3D model. Here we paid specific attention to clarifying why and how the Bern3D model was used.*

*Changes in the manuscript: Addition of a new methods section 2.5 and Appendix A on the Bern3D model.*

Lines 12-29: this is again a somewhat self-standing discussion part that should perhaps be a subsection.

*Author response: As described in our response to the general comment on the result and discussions section, we included a separate discussion section in the revised version of the manuscript. The second part of this new section is dedicated to discussing the remaining model-proxy data error after adjustment of the efficiency of the biological pump, which is base on the original p. 12 l. 12-29.*

*Changes in the manuscript: We restructured the text to include a new discussion section, where p. 12 l. 12-29 forms a paragraph that discusses the remaining model-proxy data error after adjustment of the efficiency of the biological pump.*

Here, and/or earlier, you should discuss the results of Odalen et al. (2019). Actually, would it be feasible to test their variable C/P ratio in your simulations?

*Author response: We assume that the reviewer refers to the paper Ödalen et al. (2020). Their results, which showed decreased $\delta^{13}C$ while keeping (regenerated) $PO_4$ constant, could indeed be included in our discussion on the remaining proxy-data mismatch. With regard to the feasibility to test variable C:P ratios in our model setup: Our model is computationally more demanding than the cGENIE model as employed by Ödalen et al (2020) and we do not have the resources for repeating long runs for this currently.*

*Changes in the manuscript: Discuss the results on variations in the C:P ratio by Ödalen et al. (2020) and their potential implications for our remaining model-data mismatch.*

Also could you quantify the dependence of your results to your model initial state, as discussed in Odalen et al. (2018)? [this reference is already cited in the manuscript].

*Author response: We did not carry out experiments with vastly different initial states. It is long known that different initial conditions for temperature and salinity can result in different circulation modes. However, in our case we assume that the initial conditions for the glacial ocean circulation would not be too different from preindustrial conditions and not fully different. Due to the high computational demand of our model, we cannot carry out multiple spin-ups (as 10,000 years done in the cGENIE model) with different initial conditions or tunings. This would not be feasible given currently available computational resources.*

*Changes in the manuscript: No changes were made.*

*Conclusions*

Add a reference to Rae et al. (2019) when discussing the importance of southern-sourced waters. This should probably also be discussed earlier in the results/discussion.

*Author response: We assume that the reviewer refers to the paper Rae et al. (2018). As our paper does not deal with pH changes, we do not specifically discuss this article, but can include this reference as it highlights the central role of SSW as the reviewer points out.*

*Changes in the manuscript: Rae et al. (2018) is now cited on page 2 where the importance of SSW is mentioned, and is added to the reference list accordingly.*

*Technical corrections*

Abstract Line 17: ocean model state? Do you mean "equilibrium simulations"? Clarify. Ocean model state is not the best term to use here.

*Author response: We adjusted the text as described below*

*Changes in the manuscript: We replaced the sentence 'We prepared a PI and LGM ocean model state (NorESM-OC) with full biogeochemistry (including the carbon isotopes δ13C and radiocarbon) and dynamic sea ice.' with 'We prepared a PI and LGM equilibrium simulation using the ocean model NorESM-OC with full biogeochemistry (including the carbon isotopes δ13C and radiocarbon) and dynamic sea ice.'*

Line 23: "we explore the theoretical effects" doesn't quite make sense. This could just be "we explore/test the effects".

*Author response: We think clarifying that our approach is exploring the potential effects only (i.e. it is an approximation as no actual simulation is done) is important here, and we therefore replace 'theoretical' with 'potential (offline)' in the abstract.*

*Changes in the manuscript: Replaced 'theoretical' with 'potential (offline)' on p.1 l.23.*

Line 29: again "theoretical" is not quite the right word. Just say "our approach". Same in the rest of the manuscript (e.g. page 7, 10, 13). Perhaps do just call it "offline".

*Author response: See our response to the previous comment.*

*Changes in the manuscript: We clarified the use of the word 'theoretical' throughout the text by adding the word '(offline)' or 'potential' to clarify our intention to explore the potential (offline) effects whenever we describe our approach.*

Page 10, line 30: miss-match should be mismatch.

*Author response: Thank you for noting this mistake, we will adjust the manuscript as suggested.*

*Changes in the manuscript: change p. 10, l. 30 miss-match to mismatch.*

Everywhere: "Southern Source" should really be "southern-sourced".

*Author response: We revisited the literature and see that both southern source water (e.g., Adkins, 2013; Curry and Oppo, 2005; Roberts et al., 2010) and southern-sourced water (Howe et al., 2016; Pöppelmeier et al., 2018) are commonly used. We therefore feel the current use of Southern Source Water (SSW) throughout the manuscript can be maintained.*

*Changes in the manuscript: None.*

References

Burke, A., Stewart, A.L., Adkins, J.F., Ferrari, R., Jansen, M.F. and Thompson, A.F., 2015. The glacial middepth radiocarbon bulge and its implications for the overturning circulation. Paleoceanography, 30(7), pp.1021-1039.

Klockmann, M., Mikolajewicz, U. and Marotzke, J., 2016. The effect of greenhouse gas concentrations and ice sheets on the glacial AMOC in a coupled climate model. Climate of the Past, 12, pp.1829-1846.

Marzocchi, A. and Jansen, M.F., 2019. Global cooling linked to increased glacial carbon storage via changes in Antarctic sea ice. Nature Geoscience, 12(12), pp.1001.

Ödalen, M., Nycander, J., Ridgwell, A., Oliver, K.I., Peterson, C.D. and Nilsson, J., 2019. Variable C/P composition of organic production and its effect on ocean carbon storage in glacial model simulations. Biogeosciences Discussions, pp.1-33. (accepted) DOI: https://doi.org/10.5194/bg-2019-149

Stein, K., Timmermann, A., Kwon, E.Y. and Friedrich, T., 2020. Timing and magnitude of Southern Ocean sea ice/carbon cycle feedbacks. Proceedings of the National Academy of Sciences, 117(9), pp.4498-4504

*References of the response*

*Adkins, J. F.: The role of deep ocean circulation in setting glacial climates, Paleoceanography, 28, 539-561, 10.1002/palo.20046, 2013.*

Burke, A., Stewart, A. L., Adkins, J. F., Ferrari, R., Jansen, M. F., and Thompson, A. F.: The glacial mid-depth radiocarbon bulge and its implications for the overturning circulation, Paleoceanography, 30, 1021-1039, 10.1002/2015PA002778, 2015.

Curry, W. B., and Oppo, D. W.: Glacial water mass geometry and the distribution of δ13C of ΣCO2 in the western Atlantic Ocean, Paleoceanography, 20, 10.1029/2004PA001021, 2005.

Galbraith, E. D., and Skinner, L. C.: The Biological Pump During the Last Glacial Maximum, Annual Review of Marine Science, 12, 559-586, 10.1146/annurev-marine-010419-010906, 2020.

Klockmann, M., Mikolajewicz, U., and Marotzke, J.: The effect of greenhouse gas concentrations and ice sheets on the glacial AMOC in a coupled climate model, Clim. Past, 12, 1829-1846, 10.5194/cp-12-1829-2016, 2016.

Roberts, N. L., Piotrowski, A. M., McManus, J. F., and Keigwin, L. D.: Synchronous Deglacial Overturning and Water Mass Source Changes, Science, 327, 75, 10.1126/science.1178068, 2010.

Howe, J. N. W., Piotrowski, A. M., Noble, T. L., Mulitza, S., Chiessi, C. M., and Bayon, G.: North Atlantic Deep Water Production during the Last Glacial Maximum, Nature Communications, 7, 11765, 10.1038/ncomms11765, 2016.

Jansen, M. F.: Glacial ocean circulation and stratification explained by reduced atmospheric temperature, Proceedings of the National Academy of Sciences, 114, 45-50, 10.1073/pnas.1610438113, 2017.

Marzocchi, A., and Jansen, M. F.: Global cooling linked to increased glacial carbon storage via changes in Antarctic sea ice, Nature Geoscience, 12, 1001-1005, 10.1038/s41561-019-0466-8, 2019.

Pöppelmeier, F., Gutjahr, M., Blaser, P., Keigwin, L. D., and Lippold, J.: Origin of Abyssal NW Atlantic Water Masses Since the Last Glacial Maximum, Paleoceanography and Paleoclimatology, 33, 530-543, 10.1029/2017PA003290, 2018.

Rae, J. W. B., Burke, A., Robinson, L. F., Adkins, J. F., Chen, T., Cole, C., Greenop, R., Li, T., Littley, E. F. M., Nita, D. C., Stewart, J. A., and Taylor, B. J.: CO2 storage and release in the deep Southern Ocean on millennial to centennial timescales, Nature, 562, 569-573, 10.1038/s41586-018-0614-0, 2018.

Stein, K., Timmermann, A., Kwon, E. Y., and Friedrich, T.: Timing and magnitude of Southern Ocean sea ice/carbon cycle feedbacks, Proceedings of the National Academy of Sciences, 117, 4498, 10.1073/pnas.1908670117, 2020.

*Ödalen, M., Nycander, J., Ridgwell, A., Oliver, K. I. C., Peterson, C. D., and Nilsson, J.: Variable C/P composition of organic production and its effect on ocean carbon storage in glacial-like model simulations, Biogeosciences, 17, 2219–2244, https://doi.org/10.5194/bg-17-2219-2020, 2020.*

---

## Author Response (AR2)

Dear Editor Laurie Menviel, Dear anonymous Referee #1 and #2,

Thank you for your suggestions and comments on our revised manuscript from the 12[th] of December 2020. We have replied to your comments below (in italics), in the following order: 1) the comments from the Editor, 2) the comments from Referee #1, 3) the comments from Referee #2.

Yours sincerely,

Anne Morée and co-authors
* * *
**Editor Comments**

1) The simulated changes in oceanic circulation in the Atlantic Ocean seem relatively small, and there is no shoaling (or even a slight deepening) of NADW at the LGM compared to PI. The LGM ACC is stronger, does that mean the Southern Ocean upwelling is stronger at the LGM? In any case, given the text in section 3.2.1, it should be made clearer in the Abstract, Discussion and Conclusion, that stronger changes in oceanic circulation could also provide a better model-data agreement, and given little changes in oceanic circulation a Beff of 75% is needed.

*The LGM-PI changes in ocean circulation are visualized in Fig. S6 (Atlantic stream function), as well as through the PO tracer Fig. 1a,b and Fig. S11a,b. Based on the Atlantic stream function, we find a 350m shoaling of the 0 Sv contour line at 30°S, and based on the PO tracer we see a more voluminous and northward reaching SSW mass. The ACC has strengthened by about 13% LGM-PI. The latter indeed goes along with a strengthened upwelling south of ~55 °S (see Fig 1 below).*

[Figure]

*Figure 1 Global climatological mean stream function, difference between LGM-PI.*

*The LGM-PI changes in ocean are generally within the uncertainties of reconstructions (as described in detail in Section 3.2.1). We nevertheless agree that there is additional room for changes in ocean circulation that could improve the d13C model-proxy data agreement (as discussed in Sect 4.1). A reduction instead of an increase in Southern Ocean upwelling between the LGM and PI would be one candidate for improvement. Note however, that this does not contradict our approach to explore the BP_eff as a strengthening of the efficiency of the biological pump can also be obtained through 'pure' circulation changes as this are inherently related to the biological pump (see also the last sentence of Sect 2.4). Nevertheless, the doubling of the strength of the efficiency of the biological pump still leaves a model-proxy data error of 0.07 permil larger than the 0.19 permil data uncertainty, which can likely be improved through further changes in ocean dynamics (last sentences Abstract). In order to put more*

*stress on and clarify the possibility of ocean circulation changes reducing the model-proxy data error, we:*

*- included the sentence 'The drivers of such an increase in the biological pump efficiency may be both biological as well as related to circulation changes incompletely captured by our model - such as stronger isolation of Southern Source Waters.' in the Abstract;*

*- include the sentences 'Last, we note that many of the biogeochemical processes mentioned here are closely related to ocean circulation. Therefore, changes in LGM-PI water mass configuration and overturning strength beyond those captured by our model are strong candidates for reducing model-proxy data biases.' At the end of (the new) Sect 4.2.;*

*- including the sentence 'This approximate doubling is likely driven by a combination of additional biological and physical changes, such as stronger isolation of SSW (as discussed in detail in Sect 4.1 and 4.2).' in the conclusion.;*

*-We add the sentence 'The strengthening of the ACC in our simulation goes along with a strengthening in upwelling south of ~55 °S (not shown)' to Sect. 3.2.1 to clarify the effect of the ACC strengthening on upwelling in our LGM simulation.*

2) Some clarifications might also be needed regarding the LGM to PI shift in mean d13C: the mean oceanic d13C was about 0.34 permil lower at the LGM than PI (e.g. Peterson et al., 2014). Given that the LGM and PI atmospheric d13CO2 are relatively similar, this shift arises from a lower terrestrial carbon reservoir at the LGM, and potential imbalances between weathering and sedimentation. As the LGM oceanic state is forced with -6.5 permil d13CO2, this shift is not included, which is fine. However, caution has to be taken while comparing model and data d13C, as the model values will of course be higher than the data. I am saying this because I am worried that when adjusting the Beff, this was not taken into account, which would lead to an "over-adjustment" of the Beff.

*We note there is some confusion regarding the setup of our atmospheric carbon isotopes. Our LGM oceanic state is not forced by a -6.5% permil $\delta^{13}C^{atm}$, instead the box atmosphere lets it freely evolve. We include a sentence at the end of Sect 2.2 'Note that atmospheric $\delta^{13}C$ can freely evolve in our setup due to the inclusion of a prognostic box atmosphere.' as well as including '…, after which these are allowed to freely evolve,' in Sect. 2.3 in order to make this clearer.*

*The effect on atmospheric and marine $\delta^{13}C$ due to vegetation loss on land is indeed not included in our study (as stressed in Sect 3.2.2). Nevertheless, we see we have not reported the actual LGM-PI atmospheric $\delta^{13}C$ change as simulated by our model nor the change in mean marine $\delta^{13}C$. Mean marine $\delta^{13}C$ changes +0.2148 ‰ (from 0.5403 ‰ in the PI to 0.7551 ‰ in the LGM). We have added these numbers to Sect 3.2.2, as well as for atmospheric d13C.*

*Indeed, if our model would have contained the input of low $\delta^{13}C$ from the land to the atmosphere and ocean, this may have contributed to the mean marine $\delta^{13}C$ shift. We think that the mean marine shift of 0.34 ± 0.19 ‰ as estimated by Peterson et al., (2014) is driven not only by lower terrestrial carbon reservoir at the LGM and potential imbalances between weathering and sedimentation (and the atmospheric shift which is small): The stronger LGM marine $\delta^{13}C$ gradient represents a voluminous deep ocean with a negative signature, thereby contributing to lower mean marine $\delta^{13}C$. The effect of sedimentation-weathering imbalances plus lower terrestrial carbon on mean marine $\delta^{13}C$ was estimated in Jeltsch-Thömmes et al. (2019) at a relatively uniform deglacial change of ~0.4 permil (i.e.*

*-0.4 permil LGM-PI) for a change in the land biosphere carbon inventory of 890 Gt C – within the uncertainty range of the Peterson et al. (2014) estimate. Nevertheless, the 0.4 ‰ mean $\delta^{13}C$ shift also leads to a change in $\delta^{13}C^{atm}$ of similar size (Jeltsch-Thömmes et al., 2019), which is not seen in e.g. ice core records. Several processes thus need to be considered simultaneously to estimate their effects on $\delta^{13}C^{atm}$ and $\delta^{13}C\_DIC$ – and no consensus is reached here yet. Our model simulated a ~0.2 ‰ increase in LGM-PI mean $\delta^{13}C$ in absence of some of these processes (most importantly, a land and sediment model). In summary, we note that the quantitative contribution of each of the drivers of the shift in mean marine $\delta^{13}C$ is still under debate and that our model setup only captures some of these drivers (as no land or sediment model is included). We agree that it is important for the reader to get an impression of the potential effects of a mean $\delta^{13}C$ shift on our analysis (as it indeed can cause an overestimation of LGM BP\_eff). Therefore, we decided to include an estimate of the potential effects of a shift in mean d13C on our analysis: If we redo the analysis done to make Fig. 5 but including a -0.4 (or -0.2 permil) shift in the LGM results before comparing to the proxy data, the Figure would look as in the new Fig. S19, with a best-fit BP\_eff of ~55 % (~65%). We included a new paragraph on this in the end of Sect 4.4 as well as in the Summary and Conclusions to discuss these points.*

*Besides the above:*
*- While adding the data on the atmospheric carbon isotopes to the manuscript, we noted two small errors in the reporting of the LGM-PI pCO2 changes as well, which is 20.3 ppm exactly (corrected in the revision).*
*- For radiocarbon, all data are calibrated after the model simulation to an atmospheric value of 0 ‰ (last sentence Sect 2.1) to facilitate comparison with other studies.*
*- We included the words 'up to' in front of the 75 % when describing the conclusion of our analysis on BP\_eff where appropriate throughout the text.*

3) Previous LGM model-data comparisons, and particularly the ones using d13C, are completely ignored (e.g. Tagliabue et al. 2009; Hesse et al., 2011; Gebbie 2014; Schmittner & Somes, 2016 (this one is discussed a bit); Menviel et al., 2017, Muglia et al., 2018; Menviel et al., 2020). A paragraph in the Introduction would have been nice, but a few sentences putting your results in the context of these studies definitely need to be added in the discussion.

*We had included and discussed the papers by Gebbie (2014) (p. 8, l. 36; p. 9, l. 9; p. 10, l. 27; ), Schmittner and Somes (2016) (p. 15, l. 2-9), Menviel et al. (2017) (p. 5, l. 8), and Muglia et al. (2018) (p. 5, l. 13; p. 9, l. 1) in our revised version together with other publications on LGM model-data comparisons. It is unclear to us how the impression could arise that these "are completely ignored". In order to extend the discussion with respect to previous work, we have included further text sections and references in the introduction and discussion (See changes in Introduction Sect. 1 second paragraph, beginning Sect. 4 and Sect. 4.3).*

**Referee #1**

Line16 (Abstract) : "presents a realization of" --> "discusses"
*Author Response and change in the manuscript: Corrected as suggested*

Line23 (Abstract) : " (offline)" --> removed; The meaning of the word "(offline)" is not very clear.

*Author Response and change in the manuscript: Removed as suggested. We also included an explanation of the word offline in Sect. 2.4 and made small adjustment throughout the manuscript where the word 'offline' is used.*

Line28 (Abstract): ", on which we include a detailed discussion. " --> removed (or the content of discussion should be described explicitly)
*Author Response and change in the manuscript: Removed as suggested. We have also rewritten the last few sentences of the Abstract to better stress the potential role of ocean circulation changes in improving the model-proxy data bias.*

**Referee #2**

Abstract: remove the last statement "on which we include a detailed discussion" as it is unnecessary here.
*Author Response and change in the manuscript: Removed as suggested. We have also rewritten the last few sentences of the Abstract to better stress the potential role of ocean circulation changes in improving the model-proxy data bias.*

Page 5, lines 30-31: it would be appropriate to cite at least one earlier reference and especially one that is based on the analysis of the sedimentary record, rather than only numerical simulations - e.g. Kohfeld et al. (2005). Kohfeld, K.E., Le Quéré, C., Harrison, S.P. and Anderson, R.F., 2005. Role of marine biology in glacial-interglacial CO2 cycles. Science, 308(5718), pp.74-78.
*Author Response and change in the manuscript: We agree that there are certainly other references that could be listed here. We included several more from both modelling- and sedimentary record-based reconstructions.*

Page 7, line 20: I still don't find "theoretical approach" the right definition here and a somewhat confusing way to call this. These are "offline calculations of the biological pump efficiency" or something along these lines.
*Author Response and change in the manuscript: We have replaced this sentence with your suggestion. Additionally we made revisions throughout the text where the use of the terms 'offline' or 'theoretical approach' may cause confusion.*

Page 7, lines 30-32: this sentence is a little long. Break up or at least add a comma between "mechanisms" and "obtained".
*Author Response and change in the manuscript: We broke up the sentence in two and made some small adjustments to improve readability.*

Page 8, line 1: figure should be sequential, but here you reference figure 7. But this is probably fine if you at least state that this figure is in Appendix A (here and everywhere else in the text afterward). That said, I am not sure that you need both a Supplementary Material and an Appendix?
*Author Response and change in the manuscript: We have included 'Appendix A' to every mention of Fig. 7. We like to keep both a SM and Appenix on the Bern3D model as otherwise we find the main text becomes too long. Details are now in the SM, while more general information on the approach and model are in the Appendix and main text.*

Page 11, line 12: I just find this reference to a "theoretical framework" unnecessarily confusing again here. All you need to say really is something like: […] and discuss the LGM-PI changes by exploring the efficiency of the biological pump (Sect. 3.3). Similarly in other parts of the text, as noted before.
*Author Response and change in the manuscript:* We assume the reviewer means page 8 here. See also our reply to your comment on Page 7, line 20.

Page 9, Line 28: "weaker" rather than "slower".
*Author Response and change in the manuscript: Changed as suggested*

Page 10, line 3: the meaning of the first part of the sentence ("Besides radiocarbon aging of in particular SSW") is unclear, so please rephrase.

*Author Response and change in the manuscript: We shortened this first part of the sentence to improve clarity, and added the reference to the relevant Atlantic section figure (Fig. 1c)*

I think that supplementary Figure S2 is useful, but it doesn't seem to be referenced anywhere in the text.
*Author Response and change in the manuscript: Fig. S2 is only referred to in SM text 1, as is Fig. S1. We have now added a direct reference to Fig. S2 in our reference to SM text 1 (in Sect. 2.2).*

Page 15, line 7: as well as
*Author Response and change in the manuscript: Thank you for noting this mistake, we added 'as' here.*

Page 14 onwards: the current "Discussion" session (4) is clearer as a self-standing section, but a little heavy as it is and could use some breaking up into subsections from line 15 onwards. I would suggest, for instance, renaming the whole section something like "LGM model-data biases" and add a couple more subsections to break this up a little. So the current subsection 4.2 could keep the same title but become 4.4 and before that the physical/water mass changes could be highlighted in one subsection (4.1) and then separate the biological mechanisms and DIC discussion from page 15 line 8 (4.2). Or split it in a different way if you think it is more appropriate, but add at least two additional break points here. I am saying this because this discussion is interesting and useful in highlighting several important results from these simulations, but it is hard to follow and the reader would struggle with it as is.
*Author Response and change in the manuscript: Thank you for your suggestions to improve the readability of this discussion section. We now have 4 subsections none of which are more than a page long, which we hope helps the reader navigate through this discussion.*

Page 18, line 5 onwards: it may be useful to refer to, as another example, this newly accepted paper: Zhu et al. (2021), GRL, Assessment of equilibrium climate sensitivity of the Community Earth System Model version 2 through simulation of the Last Glacial Maximum. Where the LGM simulation is used to show biases that would also affect simulations used for future projections with the same model (even though the focus is on equilibrium climate sensitivity and cloud feedbacks).
*Author Response and change in the manuscript: Thank you for pointing us to this new article. We have included its reference here.*